# Afadin-deficient mouse retinas exhibit severe neuronal lamination defects but preserve visual functions

Akiko Ueno[1,2,3†], Konan Sakuta[4†], Hiroki Ono[2], Aki Hashio[4], Haruki Tokumoto[4], Mikiya Watanabe[4], Taketo Nishimoto[4], Toru Konishi[2], Yuki Emori[4], Shunsuke Mizuno[4], Mao Hiratsuka[4], Jun Miyoshi[5], Yoshimi Takai[6], Masao Tachibana[1], Chieko Koike[1,2,3]*

[1]Center for Systems Vision Science, Research Organization of Science and Technology, Ritsumeikan University, Kyoto, Japan; [2]College of Pharmaceutical Sciences, Ritsumeikan University, Kyoto, Japan; [3]Ritsumeikan Global Innovation Research Organization (R-GIRO), Ritsumeikan University, Kyoto, Japan; [4]Graduate School of Pharmacy, Ritsumeikan University, Kyoto, Japan; [5]Department of Molecular Biology, Osaka Medical Center for Cancer and Cardiovascular Disease, Osaka, Japan; [6]Division of Pathogenetic Signaling, Department of Physiology and Cell Biology, Kobe University Graduate School of Medicine, Kobe, Japan

*For correspondence: koike@fc.ritsumei.ac.jp

†These authors contributed equally to this work

## eLife Assessment

This study demonstrates that conditional knockout of afadin disrupts retinal laminar organization and reduces the number of photoreceptors, while preserving certain aspects of retinal ganglion cell structure and light responsiveness. The work is **valuable** and well-supported by revised figures and comprehensive data on retinal cell types, lamination patterns, and visual functio. The findings are **solid** and intriguing, and the study provides insights into the relationship between retinal lamination and neural circuit function.

**Abstract** Neural lamination is a common feature of the CNS, with several subcellular structures, such as adherens junctions (AJs), playing a role in this process. The retina is also heavily laminated, but it remains unclear how laminar formation impacts retinal cell morphology, synapse integrity, and overall retinal function. In this study, we demonstrate that the loss of afadin, a key component of AJs, in mice leads to significant pathological changes. These include the disruption of outer retinal lamination and a notable decrease as well as mislocalization of photoreceptors, their outer segments, and photoreceptor synapses. Interestingly, despite these severe impairments, we recorded small local field potentials, including the a- and b-waves. We also classified retinal ganglion cells (RGCs) into ON, ON-OFF, and OFF types based on their firing patterns in response to light stimuli. Additionally, we successfully characterized the receptive fields of certain RGCs. Overall, these findings provide evidence that retinal circuit function can be partially preserved even when there are significant disruptions in both retinal lamination and photoreceptor synapses. Our results indicate that retinas with severely altered morphology still retain some capacity to process light stimuli.

## Introduction

A common feature of the CNS in vertebrates is highly laminated structures, which are established by well-ordered neuronal migration (*Ayala et al., 2007*; *Liu, 2011*; *Veeraval et al., 2020*). Lamination defects are observed in some patients with psychiatric disorders, such as autism spectrum disorder, and these rodent models suggest that lamination may be associated with neural circuit formation and function (*Miao et al., 1994*; *Pan et al., 2019*; *Romero et al., 2018*; *Stouffer et al., 2016*). However, several studies report that orderly lamination is dispensable for the assembly of direction-selective tectal circuits as well as for several cortical circuit functions and synaptic connections (*Dräger, 1981*; *Guy et al., 2015*; *Guy and Staiger, 2017*; *Nikolaou and Meyer, 2015*). Therefore, the full extent of the association between lamination and the integrity of neural circuit function remains elusive.

Adherens junctions (AJs), known to be correlated with CNS lamination, are adhesive intercellular junctions composed of protein complexes that mediate strong cell-cell adhesion (*Harris and Tepass, 2010*; *Masai et al., 2003*; *Meng and Takeichi, 2009*; *Park et al., 2002*; *Takai et al., 2008b*). Cadherin-catenin complex, nectin-afadin complex, and actin cytoskeleton are the major components of AJs (*Gil-Sanz et al., 2014*; *Harris and Tepass, 2010*; *Ikeda et al., 1999*; *Meng and Takeichi, 2009*; *Tachibana et al., 2000*; *Takai et al., 2008a*; *Takai et al., 2008b*). Cell-cell adhesion of AJs is mainly mediated by the cadherin-catenin complex, whereas nectin and afadin are required for the initial establishment of AJs (*Takai and Nakanishi, 2003*). A recent report shows that afadin in intestinal cells is essential for epithelial cell-cell adhesion, which was previously assumed to be primarily regulated by cadherin and catenin (*Mangeol et al., 2024*). Following this result, the functional importance of afadin and nectin in AJs is being reconsidered (*Sebbagh and Schwartz, 2024*). Animals with mutations in AJ molecules exhibit cell migration defects in the developing stage, resulting in disruption of lamination, which might be associated with brain dysfunctions, such as intellectual disability (*Baum and Georgiou, 2011*; *de Ligt et al., 2012*). However, how the defects in AJs affect the CNS function also remains unclear.

The retina, one of the best-understood mammalian neural circuits in the CNS, is highly laminated and organized into three nuclear layers and two synaptic layers (*Amini et al., 2017*). In the retina, incoming light is converted into neural signals in photoreceptors, processed in parallel through a network of interneurons, and finally sent to the brain via the optic nerve, the bundle of retinal ganglion cell (RGC) axons (*Masland, 2012*). Photoreceptors, bipolar cells (BCs), and RGCs are connected sequentially, and horizontal cells and amacrine cells (ACs) modulate the information flow in the circuit (*Wässle, 2004*). The mechanisms underlying retinal circuit formation and function have been actively studied, but the effects of retinal lamination disruption on retinal neural circuits remain unclear (*Duan et al., 2018*; *Duan et al., 2014*; *Gollisch and Meister, 2010*; *Zapp et al., 2022*). In mammals, most of the major AJ molecules belong to protein families, possibly resulting in functional redundancy with no apparent lamination phenotype in a single knockout (KO) of a member of this complex. Indeed, the loss of central AJ molecules, nectin-1 and nectin-3, does not cause severe defects in retinal lamination (*Inagaki et al., 2005*). To clarify the effects of AJ molecule depletion on retinal neural circuits and cell morphology while avoiding the issue of functional redundancy, we focused on afadin, which is a scaffolding protein for nectin and has no paralog in mice. Afadin regulates the localization of nectin, which initiates cell–cell adhesion and promotes AJ formation by recruiting the cadherin–catenin complex. (*Ohama et al., 2018*; *Takai and Nakanishi, 2003*). In addition, afadin interacts with various cell adhesion and signaling molecules, as well as the actin cytoskeleton, and contributes to the accumulation of β-catenin, αE-catenin, and E-cadherin at AJs (*Sakakibara et al., 2018*; *Sato et al., 2006*). *Afadin* KO mice exhibit severe disruption of AJs in the ectoderm, along with other developmental defects, leading to embryonic lethality (*Ikeda et al., 1999*; *Zhadanov et al., 1999*). Conditional deletion of afadin in RGCs leads to disruption of dendrites in ON-OFF direction-selective RGCs (*Duan et al., 2018*). However, the effect of afadin loss on retinal lamination, circuit formation, and function is poorly understood.

In this study, using the *Afadin* conditional KO (cKO) mouse, we revealed that the *Afadin*-deficient retinas exhibit severe pathological defects, such as outer retinal lamination disruption, as well as decrease and mislocalization of photoreceptors and their synapses. In contrast to severe disruption of the outer retina, the inner plexiform layer (IPL) structure in the inner retina was relatively intact. Using the retinas isolated from the *Afadin* cKO mouse, we could record local field potentials (micro electoretinograms: mERGs) and RGC firings. Based on the light-evoked firing pattern, RGCs could

be classified into ON, ON-OFF, OFF, and other types as previously established. Furthermore, the receptive field (RF) could be mapped in some RGCs. Consistent with this, the cKO mice responded to moving stimuli in visual behavioral tests. Our results suggest that neural circuits in the retina with severe defects of the outer retinal lamination and photoreceptor synapses can mediate some visual information processing and partially preserve visual functions.

## Results

### Afadin is localized to adherens junctions, synaptic regions, and the surface of bipolar cells in the retina

To elucidate afadin localization in mature retina (1-month-old: 1 M), we immunostained retinal sections with anti-afadin antibody. Afadin signals were observed in the outer segment (OS), outer limiting membrane (OLM), outer plexiform layer (OPL), inner nuclear layer (INL), and IPL (*Figure 1A Top*). These signals were not observed in retinal sections immunostained using anti-afadin antibody pre-absorbed with the antigen peptide (*Figure 1A Bottom*). Thus, we confirmed that the signals are specific to afadin. To explore whether afadin is co-localized with nectin in the retina, we analyzed afadin and nectin localization in the retina by immunostaining using antibodies against afadin and nectins 1-3. Afadin signals were aligned and overlapped with all three nectin signals in the OLM (*Figure 1B, C*, *Figure 1—figure supplement 1A*), consistent with the previous finding that AJs are formed in the OLM in the retina (*Willbold and Layer, 1998*). Nectin-1, unlike nectin-2 and nectin-3, was partially co-localized with afadin at the OPL and IPL, in addition to the OLM (*Figure 1B*, *Figure 1—figure supplement 1B*). Nectin-1 was co-localized with afadin in cone synapses immunostained with Arrestin3 (Arr3, cone photoreceptor marker), suggesting that puncta adherent junctions may be formed between cone pedicles and BC dendritic tips. In the retina at postnatal day 0 (P0), afadin and nectin-1 signals were observed in the outer retinal surface where AJs are formed (*Figure 1—figure supplement 1C*; *Koike et al., 2005*). Immunostaining with anti-afadin, anti-SCGN (Type 2–6 cone BC marker; *Puthussery et al., 2010*), and anti-PKCα (rod BC marker) revealed that afadin localizes to the dendritic tips, cell bodies, and axons of ON-BCs and at least cell bodies of OFF-BCs (*Figure 1D*).

### Afadin is required for proper retinal lamination

To investigate the role of afadin in the retina, we generated retina-specific *Afadin* cKO mice by crossing *Afdn^flox^* mice with *Dkk3-Cre* mice (*Majima et al., 2009*; *Sato et al., 2007*; *Yamamoto et al., 2020*). Cre activity in *Dkk3-Cre* mice was confirmed by crossing with R26R-H2B-EGFP reporter mice (*Abe et al., 2011*), in which Cre-dependent excision of a *loxP*-flanked STOP cassette drives nuclear EGFP expression from the *Rosa26* locus (*Figure 2—figure supplement 1A*, B). Afadin was not detected at the protein level by western blotting in the *Afadin* cKO retina (*Figure 2—figure supplement 1C*). The *Afadin* cKO mouse was viable and showed no gross morphological abnormalities. To examine retinal lamination and morphology, using the cKO retina at 1 M, we performed toluidine blue staining. We found that there were no clear outer and inner segments of photoreceptors in the outer retina, no distinct OPL, and no clear compartmentalization of the ONL and INL (*Figure 2A*). Rod photoreceptors, identified by inverted nuclear architecture (*Solovei et al., 2009*; *Figure 2A* insets), were widely scattered in the cKO retina, whereas the IPL and GCL appeared intact. Furthermore, rosette-like structures, which lacked cell bodies centrally, were sometimes observed in the cKO outer retina (*Figure 2A, B and C*: arrowhead).

To investigate the defects in the OS of the cKO retina in more detail, we stained retinal sections with rhodamine-labeled peanut agglutinin (PNA, a marker of cone photoreceptor OS and axon terminals) or antibodies against Rom1 (rod OS marker), Rhodopsin (rod OS marker), S-opsin (S-cone OS marker), and M-opsin (M-cone OS marker). Rhodopsin, Rom1, S-opsin, and M-opsin play central roles in phototransduction in the OS (*Clarke et al., 2000*; *Greenwald et al., 2014*; *Tanimoto et al., 2015*; *Xu et al., 2020*). Distribution of Rom1, Rhodopsin, S-opsin, M-opsin, and PNA signals in the cKO retina differed significantly from those of the *Afadin* conditional heterozygous (cHet) retina. In the cKO retina, Rom1, Rhodopsin, S-opsin, M-opsin, and PNA signals were remarkably decreased in number and mislocalized in the retina (*Figure 2B and C*). These markers were not observed in the rosette-like structure of the cKO retina, suggesting that the rosette-like structure is different from the photo-receptor rosette as reported (*Stuck et al., 2012*). Electron microscopic analysis also demonstrated

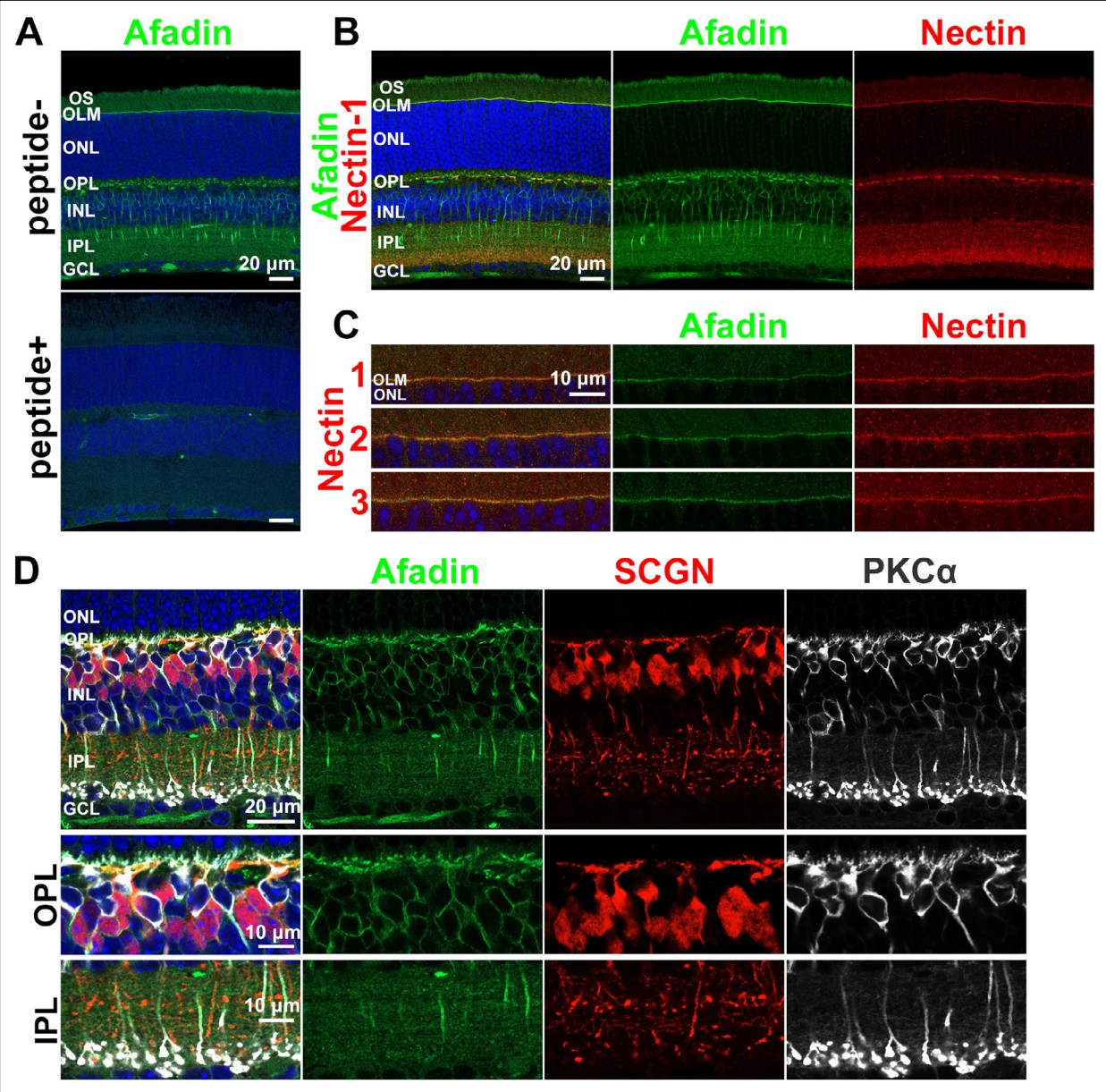

**Figure 1.** Afadin is localized to the OS, OLM, OPL, INL, and IPL. (**A**) Immunostaining of the wild-type (WT) mouse retina (1 M) with anti-afadin antibody. The afadin signals disappeared in the pre-absorbed (peptide+) sample. The signal observed in the peptide + image represents the background and non-specific staining. Nuclei were stained with DAPI (blue). (**B**) Immunostaining of the WT retinal section (1 M) using anti-afadin (green) combined with anti-Nectin-1 (red) antibody. Necin-1 was partially co-localized with afadin in the OLM, OPL, and IPL. (**C**) Afadin was colocalized with nectin-1, nectin-2, and nectin-3 at the OLM. The 1 M WT retinal sections were immunostained with anti-afadin (green), anti-nectin-1 (red, upper panels), anti-nectin-2 (red, middle panels), and anti-nectin-3 (red, lower panels) antibodies. (**D**) Immunostaining of 1 M WT retinal sections using anti-afadin (green), anti-SCGN (red), and anti-PKCα (white) antibodies. Afadin signals overlapped with SCGN and PKCα in the OPL and INL, and PKCα in the IPL. OS, the outer segment; IS, the inner segment; OLM, the outer limiting membrane; ONL, the outer nuclear layer; OPL, the outer plexiform layer; INL, the inner nuclear layer; IPL, the inner plexiform layer; GCL, the ganglion cell layer.

The online version of this article includes the following figure supplement(s) for figure 1:

**Figure supplement 1.** Afadin is localized to AJs in developing and mature retinas.

that only a few OS structures were observed between retinal cells in the disrupted ONL and INL regions of the cKO retina (*Figure 2—figure supplement 1D*). These findings indicate that rod and cone phototransduction may be severely impaired in the cKO retina. To investigate the distribution of cone and rod photoreceptors in the cKO retina, we performed whole-mount immunostaining with

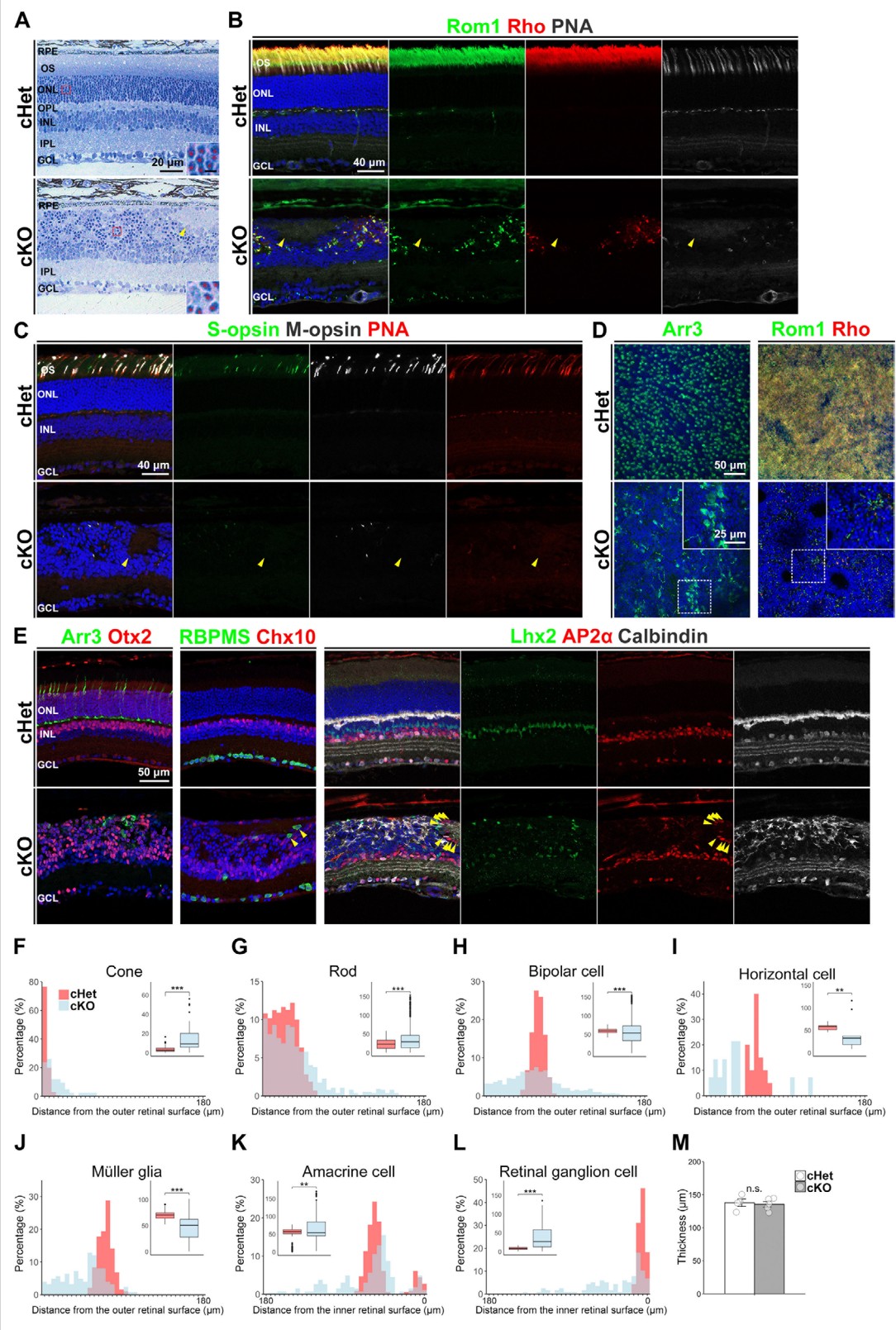

**Figure 2.** Outer retinal lamination is severely disrupted in the *Afadin* cKO retina. (**A**) Toluidine blue staining of the cHet and cKO mouse retinal sections at 1 M. The retinal layer structure was disrupted in the cKO retina. Rod photoreceptor nuclei (insets) were deeply stained. The red asterisk indicates a rod photoreceptor. The arrowhead indicates a rosette-like structure. Scale bar in the inset, 2.5 µm. (**B**) Representative images of the cHet and cKO retinas (1 M) stained with anti-Rom1 (green) and anti-Rhodopsin (Rho, red) antibodies and PNA-rhodamine (white). Rom1, Rhodopsin, and PNA signals

*Figure 2 continued on next page*

*Figure 2 continued*

were remarkably decreased and scattered in the cKO mice. The arrowhead indicates the rosette-like structure. (**C**) Representative images of the cHet and cKO retinas (1 M) stained with anti-S-opsin (green) and anti-M-opsin (white) antibodies and PNA-rhodamine (red). Cone OSs were aberrantly located in the cKO retina. The arrowhead indicates the rosette-like structure. (**D**) Retinal flat-mount immunostaining of the cHet and cKO retinas immunostained with anti-Arr3 (green, left panels), anti-Rom1 (green, right panels), and anti-Rhodopsin (red, right panels) antibodies. The inset, a magnified view of the area enclosed by the dashed box, shows an area with a relatively high density of photoreceptors in the cKO retina. (**E**) Immunofluorescent analysis of the cHet and cKO retinas (1 M) using anti-Arr3 (cone marker, green in the left panels), anti-Otx2 (photoreceptor and BC marker, red in the left panels), anti-RNA binding protein with multiple splicing (RBPMS, RGC marker, green in second left panels), anti-Chx10 (BC marker, red in second left panels), anti-Lhx2 (Müller glia marker, green in third left and third right panels), anti-AP2α (AC marker, red in third left and second right panels), and anti-Calbindin (horizontal cell and AC marker, white in third left and right panels) antibodies. Arrowheads indicate RGCs and ACs near the rosette-like structure. (**F–L**) Distribution of each retinal cell type based on their distance from the retinal surface in the cHet and cKO mice. The shortest distance between each cell body and either the outer or inner retinal surface was measured using ImageJ. For cones (**F**), rods (**G**), bipolar cells (**H**), horizontal cells (**I**), and Müller glia (**J**), the distance was measured from the outer retinal surface. For amacrine cells (**K**) and retinal ganglion cells (**L**), the distance was measured from the inner retinal surface. *Note: In K and L, the X-axis is reversed such that short distance appears on the right.* The outer retinal surface was defined as the region immediately above the somata of the outer retinal cells. Bin width was 5 µm (0–5 up to 175–180 µm). Inset box plots show the distribution of raw values, with whiskers extending to 1.5 times the interquartile range and dots indicating outliers (cHet; Cone 3.45±2.72 µm, n=141 cells from 4 mice, Rod 23.1±13.8 µm, n=3055 cells from 4 mice, Bipolar cell 59.4±6.74 µm, n=432 cells from 3 mice, Horizontal cell 57.5±6.45 µm, n=20 cells from 3 mice, Müller glial cell 69.9±7.61 µm, n=191 cells from 3 mice, Amacrine cell 51.8±18.6 µm, n=362 cells from 3 mice, Ganglion cell 8.71±3.71 µm, n=93 cells from 3 mice, cKO; Cone 14.9±14.2 µm, n=42 cells from 4 mice, Rod 35.5±30.5 µm, n=716 cells from 4 mice, Bipolar cell 56.6±32.7 µm, n=1236 cells from 5 mice, Horizontal cell 37.2±31.2 µm, n=14 cells from 4 mice, Müller glial cell 45.9±22.9 µm, n=95 cells from 4 mice, Amacrine cell 63.8±34.6 µm, n=265 cells from 4 mice, Ganglion cell 38.9±33.6 µm, n=116 cells from 5 mice). ***p<0.001, **p<0.01 by a generalized linear mixed model (GLMM) or log1p-transformation followed by a linear mixed model (LMM). Statistical tests: Cone, Rod, log1p-transformation followed by a LMM; Bipolar cell, Horizontal cell, Müller glial cell, Amacrine cell, Ganglion cell, GLMM. (**M**) The thickness of the cHet and cKO retinas at 1 M. (cHet; 137.9±11.1 µm, n=4, cKO; 135.5±9.1 µm, n=5). Error bars, mean ± SD. No significant difference was observed between the two groups (p=0.735 by Student's t-test).

The online version of this article includes the following source data and figure supplement(s) for figure 2:

**Figure supplement 1.** AJs in the OLM are disrupted in the *Afadin* cKO retina.

(**A, B**) Immunostaining of *Dkk3*-Cre; R26R-H2B-EGFP and R26R-H2B-EGFP retinas at P0 (**A**) and 1 M (**B**) using an anti-GFP antibody (green). The Cre recombinase activity was detected in the *Dkk3*-Cre; R26R-H2B-EGFP retina at both stages. (**C**) Western blot analysis of the cHet and cKO retinas using anti-afadin (upper panel) and anti-GAPDH (lower panel) antibodies. No significant afadin band was detected in the cKO retina. (**D**) Electron microscopic analysis of the cHet and cKO retinas (1 M). A few ectopic disc structures were observed in the cKO retina. The inset, a magnified view of the area enclosed by the dashed box, shows the ectopic disc structure at high magnification. The arrowhead indicates the outer segment disc structure. RPE, the retinal pigment epithelium. (**E**) Immunostaining of the cHet and cKO retinal sections at 1 M using anti-β-catenin (green, left panels), anti-nectin-1 (red, left panels), and anti-N-cadherin (green, right panels). Signals of these markers were not observed at the outer retinal surface in the cKO mice. (**F**) Immunostaining of the cHet and cKO retinal sections at 1 M using anti-glutamine synthetase (GS, green) antibody. Obvious GS signals were observed in the inner side of the retina but not at the outer retinal surface in the cKO mice. (**G**) Retinal sections from the cHet and cKO mice at 1 M were stained with an anti-CD31 antibody (red). No notable vascular differences were observed in the cKO retina.

**Figure supplement 1—source data 1.** PDF file containing original western blots *Figure 2—figure supplement 1C*.

**Figure supplement 1—source data 2.** Original files for western blot analysis displayed in *Figure 2—figure supplement 1C*.

antibodies against Arr3, Rhodopsin, and Rom1. Cell bodies of cone photoreceptors were arranged regularly in the cHet retina, whereas cone photoreceptor arrangements and morphology were significantly disordered in the cKO retina (*Figure 2D*). Rom1, Rhodopsin, and Arr3 signals were decreased but relatively dense in some areas (*Figure 2D*). These results suggest that photoreceptor distribution is disrupted, but photoreceptors remain in some areas heterogeneously.

We further investigated AJs in the cKO retina by immunostaining with OLM adherens junction markers β-catenin, N-cadherin, and nectin-1. We found that these signals were scattered in the cKO retina (*Figure 2—figure supplement 1E*). This data suggests that AJs in the OLM are disrupted in mature cKO retina.

As toluidine blue staining of the cKO retina suggested mislocalization of each cell type, we investigated the position of each retinal cell body in the cKO retina by immunostaining with retinal cell type-specific markers; Otx2 (a photoreceptor and BC marker), Chx10 (a BC marker), Arr3, Calbindin (a horizontal cell and AC marker), AP2α (an AC marker), RBPMS (a RGC marker), and Lhx2 (a Müller glial cell marker; *Figure 2E*). All retinal cell types were aligned in restricted and orderly regions in the cHet retina. In the cKO retina, the spatial organization of rod photoreceptors, BCs, cone photoreceptors, horizontal cells, and Müller glial cells appeared particularly disrupted, whereas positions of ACs and RGCs remained relatively intact, except for the immediately surrounding region of rosette-like structures. To quantitatively assess the retinal cell positioning in the cKO retina, the distance from the

soma of each cell type to the outer or inner retinal surface was measured (*Figure 2F–L*). The distance for each cell type in the cKO retina was significantly different from that in the cHet retina. In the cKO retina, BCs, horizontal cells, Müller glial cells were positioned close to the outer retinal surface, and ACs were close to the inner retinal surface, whereas photoreceptors and RGCs were positioned away from the outer and inner retinal surfaces, respectively.

Notably, BCs in the cKO retina were broadly distributed throughout the retina (*Figure 2H*). Rod photoreceptors also tended to be dispersed throughout the cKO retina. However, approximately 75% of them in the cKO retina were located within 60 μm of the outer retinal surface, which corresponds to the region where all rods are positioned in the cHet retina (*Figure 2G*). Cone photoreceptors, horizontal cells, and Müller glial cells were similarly spread, but their distribution was largely confined to the area distal to the IPL (*Figure 2F, I and J*). Although ACs and RGCs were mainly located in the inner retina, a sparse distribution was also observed throughout the retina, which may be attributed to the cells surrounding the rosette-like structure (*Figure 2K and L*). The reduction in the distance from ACs to the inner retinal surface in the cKO retina is conceivably due to a shortening of the IPL (*Figure 2K*), although there was no significant difference in retinal thickness between cHet and cKO retinas (*Figure 2M*). The cKO mice appeared to show greater inter-individual variability in retinal layer structure and cell positioning than the cHet mice.

Immunostaining with an anti-GS antibody (a Müller glial cell marker) showed that Müller glial cell morphology was affected, especially at the distal part. However, a distinct inner limiting membrane was retained in the cKO retina (*Figure 2—figure supplement 1F*). It has been reported that defects in the distal processes of Müller glia are associated with abnormal retinal vasculature (*Shen et al., 2012*). Thus, we immunostained the cKO retina with anti-CD31, a blood vessel marker, but no apparent vascular abnormalities were detected (*Figure 2—figure supplement 1G*). These results indicate that afadin is essential for normal retinal lamination.

## Ablation of afadin results in the remarkable decrease and mislocalization of photoreceptor-BC synapses

No distinct OPL structure in the cKO retina suggests that synapses between photoreceptor and BC may be disrupted. To assess their integrity, we immunostained retinal sections of the cKO mouse (1 M) with antibodies against Bassoon (a photoreceptor synaptic ribbon marker), PSD95 (a photoreceptor synaptic terminal marker), mGluR6 (an ON-BC dendritic tip marker), GluR5 (an OFF-BC dendritic tip marker), PKCα, and PKARIIβ (a type 3b OFF cone BC marker). In the cHet retina, photoreceptor synapses containing synaptic ribbons formed a row in the OPL (*Figure 3A, B*, *Figure 3—figure supplement 1A*). On the other hand, in the cKO retina, the row of photoreceptor synapses was not observed, and photoreceptor-BC synapses were remarkably reduced in number. Bassoon and mGluR6 were aberrantly distributed, and the position of the soma and dendritic tips of rod BCs was noticeably disturbed (*Figure 3A*). Only a few ribbon synapses remained (*Figure 3A* arrowheads). The 3D reconstructions of the confocal image stacks showed more clearly that photoreceptor synapses decreased in number and mislocated in the cKO retina (*Figure 3C and D*). The contact between the axon terminal of photoreceptors and the dendrite of ON- and OFF-BCs was also significantly reduced in number, suggesting that photoreceptor-BC synapses were considerably lost in the cKO retina (*Figure 3B, D*, *Figure 3—figure supplement 1A,B*). Along with these defects, GluR5 signals with a branch-like pattern were detected in the IPL of the cKO retina, although no GluR5 signals were found in the IPL of the cHet retina (*Figure 3B and D*). Photoreceptor-ON BC ribbon synapses and photoreceptor-OFF BC synapses in the outer retinal region beneath 1 mm$^2$ of the surface area of the cKO retina were significantly reduced to about 5% and 15% of those in the cHet retina, respectively (*Figure 3E, F*, *Figure 3—figure supplement 1C*). These results indicate that afadin may regulate the formation and localization of photoreceptor synapses.

## Afadin is required for proper localization of GluR5 to OFF BC dendrites

GluR5, a kainate receptor subunit that forms functional homomeric and heteromeric channels, is specifically expressed in OFF BCs and localizes to their dendrites (*Haverkamp et al., 2003*). To investigate GluR5 localization in more detail, we immunostained the cKO retina with anti-GluR5, anti-SCGN, and anti-PKCα antibodies. We found that ectopic GluR5 signals in the IPL overlapped not only with SCGN but also PKCα in the cKO retina (*Figure 4A*). These data indicate that afadin may be required

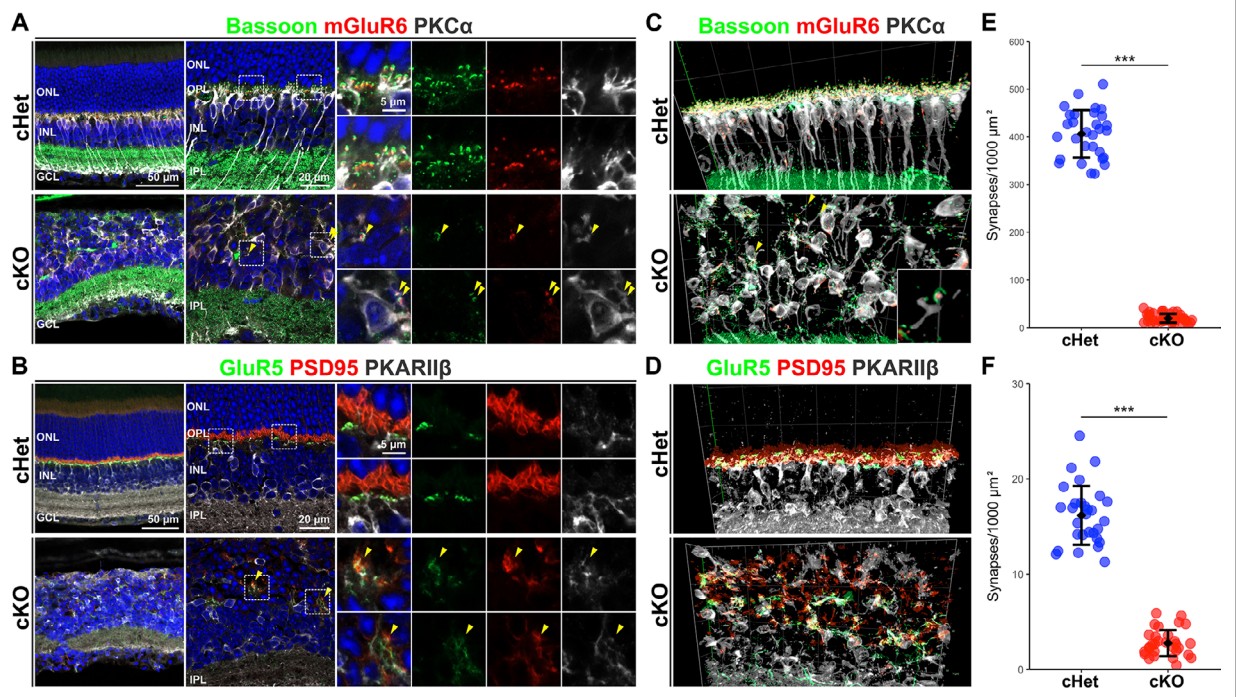

**Figure 3.** Photoreceptor-BC synapses were severely impaired and mislocalized in the *Afadin* cKO retina. (**A**) Immunostaining of the cHet and cKO retinal sections at 1 M with anti-Bassoon (green), anti-mGluR6 (red), and anti-PKCα (white) antibodies. The arrowhead indicates the synapse between rod and rod-BC. Small panels are magnified views of the area enclosed by the dashed box. (**B**) Immunostaining of the cHet and cKO retinas at 1 M with anti-GluR5 (green), anti-PSD95 (red), and anti-PKARIIβ (white) antibodies. The arrowhead indicates the synapse between the cone and OFF-BC. Small panels are magnified views of the area enclosed by the dashed box. (**C**) 3D projection of confocal image immunostaining with anti-Bassoon (green), anti-mGluR6 (red), and anti-PKCα (white) using the cHet and cKO retina (1 M). The inset, a magnified view of the area enclosed by the dashed box, shows the synapse between the rod and BC at high magnification. The arrowhead indicates the ribbon synapse between rod and rod-BC. (**D**) 3D projection of confocal image immunostaining with anti-GluR5 (green), anti-PSD95 (red), and anti-PKARIIβ (white) antibodies using the cHet and cKO retina (1 M). (**E**) Quantification of the number of synapses between rod photoreceptor and rod-BC under 1 mm$^2$ of retinal surface in the cHet and cKO mice immunostained with anti-Bassoon, anti-mGluR6, and anti-PKCα antibodies (cHet; 599.6±49.7, n=3, cKO; 29.1±12.75, n=4, 10 images from each mouse. Error bars, mean ± SD. ***p<0.001 by a GLMM). The number of synapses decreased to about 5% of the cHet in the cKO retina. (**F**) Quantification of the number of synapses between photoreceptor cells and type 3 OFF-BC under 1 mm$^2$ of the retinal surface in the cHet and cKO mice immunostained with anti-GluR5, anti-PSD95, and anti-PKARIIβ antibodies (cHet; 24.0±2.8, n=3, cKO; 4.1±1.3, n=3, 10 images from each mouse. Error bars, mean ± SD. ***p<0.001 by a GLMM). The number of synapses between the photoreceptor and Type 3 OFF-BC in the cKO retina decreased to about 15% of the cHet retina.

The online version of this article includes the following figure supplement(s) for figure 3:

**Figure supplement 1.** The OS and photoreceptor-BC synapses are affected in the developing *Afadin* cKO retina.

for specific localization of GluR5 in OFF BC dendrites. It seems likely that ectopically localized GluR5 may receive glutamate released from surrounding cells in the IPL, resulting in depolarization of ON- and OFF-BCs. It should be noted that there were no remarkable differences between cKO and cHet retinas in the localization of glutamate transporter EAAT5 and vesicular glutamate transporter vGlut1 to the axon terminal of photoreceptors and BCs (*Figure 4B*, *Figure 3—figure supplement 1D, E*). It has been reported that mGluR6 localization to ON BC dendritic tips requires trans-synaptic interaction with ELFN1 in the synaptic cleft and that ELFN1 ablation results in a remarkable reduction of mGluR6 in ON BC dendritic tips (*Cao et al., 2015*). As GluR5 localization may also be regulated by trans-synaptic interaction with photoreceptor synaptic terminal proteins, it seems possible that the defect of photoreceptor-OFF BC contacts prevents this trans-synaptic interaction, resulting in GluR5 mislocalization in the cKO retina.

Then, we investigated the IPL structure in the cKO retina by immunostaining it with anti-ChAT, Calbindin, and Calretinin antibodies. Distinct ON- and OFF ChAT bands, Calbindin bands, and Calretinin bands were observed in the cKO retina (*Figure 3—figure supplement 1F*). However, type 2–6 BCs immunostained with SCGN were slightly disturbed in the axon terminal position (*Figure 4A, B*,

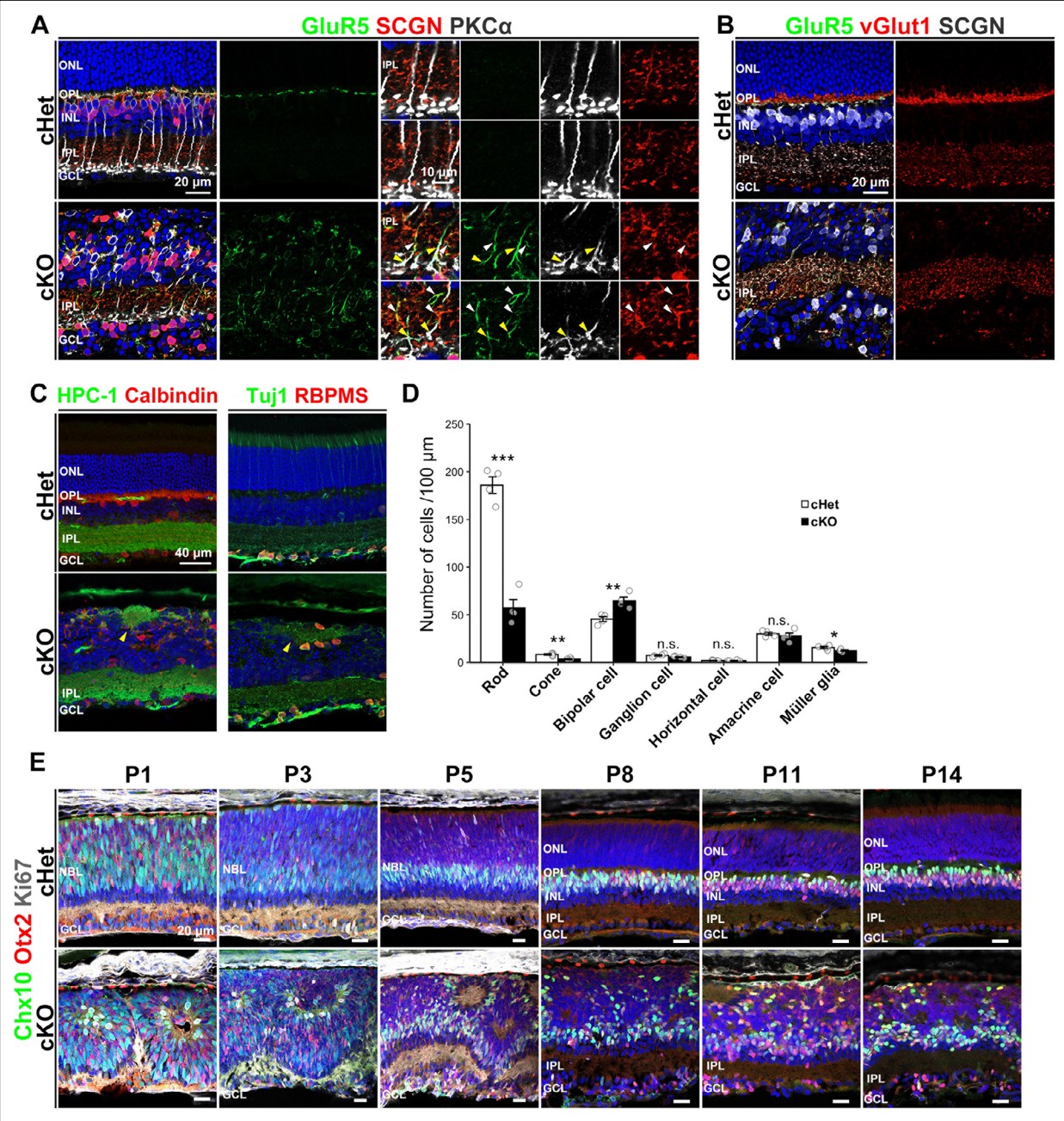

**Figure 4.** GluR5 is ectopically localized to ON and OFF BC processes. (**A**) Immunostaining of the cHet and cKO retinal sections at 1 M with anti-GluR5 (green), anti-SCGN (red), and anti-PKCα antibodies. GluR5 signal was obviously observed in the IPL of the cKO retina. Yellow and white arrowheads indicate overlap of GluR5 and PKCα, and GluR5 and SCGN, respectively. Small panels are magnified views of the area enclosed by the dashed box. (**B**) Representative images of the immunostained cHet and cKO retinas (1 M) with anti-GluR5 (green), anti-vGlut1 (red), and anti-SCGN (white) antibodies. (**C**) Immunofluorescent analysis of the cHet and cKO retinas (1 M) with anti-HPC-1 (AC marker, green in the left panels), anti-Calbindin (red in the left panels), anti-Tuj1 (RGC marker, green in the right panels), and anti-RBPMS (red in the right panels) antibodies. Arrowhead indicates acellular patches. (**D**) The number of each retinal cell type per 100 μm width of retinal section at 1 M cHet; Rod 185.9±17.5, Cone 8.4±1.4, Bipolar cell 45.5±5.2, Ganglion cell 7.4±1.5, Horizontal cell 2.1±0.4, Amacrine cell 30.1±3.2, Müller glial cell 15.7±2.3, n=4, cKO; Rod 57.3±17.3, Cone 3.5±1.3, Bipolar cell 64.6±7.8, Horizontal cell 1.9±0.7, Amacrine cell 27.6±6.3, Ganglion cell 5.7±0.9, Müller glial cell 12.4±1.5, n=4. Error bars, mean ± SD. Horizontal cell p=0.58, AC p=0.52, RGC p=0.10. ***p<0.001, **p<0.01, *p<0.05 by Student's t test. In the cKO retina, rod and cone photoreceptors and Müller glial cells significantly decreased to about 30, 40, and 80% of the cHet retina, respectively, and BCs significantly increased to about 150% of the cHet retina. (**E**) Immunofluorescent analysis of the cHet and cKO retinas at P1, P3, P5, P8, P11, and P14 using anti-Chx10 (green), anti-Otx2 (red), and anti-Ki67 (progenitor cell marker, white) antibodies. NBL, the neuroblastic layer.

The online version of this article includes the following figure supplement(s) for figure 4:

*Figure 4 continued on next page*

**Figure supplement 1.** Retinal cell fate determination is altered during development in the *Afadin* cKO mice.

**Figure supplement 2.** Retinal lamination is disputed in the *Afadin* cKO retina during development.

*Figure 3—figure supplement 1D*). These results suggest that stratification patterns in the IPL may be slightly impaired but almost intact in the cKO retina. Immunostaining analysis with anti-HPC-1 (also known as syntaxin, an AC marker), anti-Calbindin, anti-RBPMS, and anti-Tuj1 (an RGC marker) antibodies revealed that the rosette-like structure contains AC and RGC processes inside (*Figure 4C*). These data indicate that the rosette-like structure in the cKO may be an ectopic IPL, termed 'acellular patches' (*Nahar and Cho, 2022*).

## Retinal cell fate is altered in the cKO retina

To examine whether afadin loss affects retinal cell fate determination, we counted the number of retinal cells immunostained with each retinal cell marker. In the cKO retina at 1 M, BCs significantly increased to about 150% of the cHet retina, but rods, cones, and Müller glial cells were significantly decreased to about 30%, 40%, and 80% of the cHet retina, respectively (*Figure 4D*). PKCα-positive cells and SCGN-positive cells were significantly increased (*Figure 4—figure supplement 1A*), indicating that both rod- and OFF-BCs were increased in the cKO retina. In contrast, the number of rod BC processes in the IPL was significantly decreased in the cKO retina (*Figure 4—figure supplement 1B*). These data suggest that ectopically located BCs may increase in the cKO retina at 1 M. A significant increase in BCs and a significant decrease in photoreceptors were also observed in the cKO retina at P14, a stage at which retinal differentiation is complete (*Figure 4—figure supplement 1C*). Consistently, the expression of marker genes for photoreceptors and BCs was significantly decreased and increased, respectively, in the cKO retina (*Figure 4—figure supplement 1D*). In contrast, the number of Müller glial cells was unaltered in the cKO retina at P14 (*Figure 4—figure supplement 1C*), suggesting that the reduction in Müller cells observed at 1 M may result from cell death after P14 rather than a differentiation defect. To assess whether retinal cell fate changes occur during differentiation, we examined the number of BCs and the expression of rod marker genes in the cKO and cHet retinas at P1, P3, and P5. No significant differences were observed between genotypes at P1 and P3, whereas the number of BCs was significantly increased, and the expression of rod marker genes was significantly decreased in the cKO retina at P5 (*Figure 4—figure supplement 1E, F*). These results suggest that defects in cell fate determination of BCs and rod photoreceptors begin to emerge between P3 and P5. Based on these data and ectopic localization of GluR5 to ON-BCs in the cKO retina (*Figure 4A*), we propose that afadin may be required for normal retinal cell fate determination and gene expression.

## Retinal structure is severely disrupted during development in *Afadin* cKO mice

To investigate whether the aberrant lamination in the cKO retina can be attributed to a defect in development or maintenance, we immunostained the developing cHet and cKO retinas using markers for individual retinal cell types, outer segments, and photoreceptor synapses (*Figure 4E*, *Figure 4—figure supplement 2A-D*). In the cKO retina at P1, retinal cells appeared to be organized into distinct multicellular compartments with apparent boundaries (*Figure 4E*, *Figure 4—figure supplement 2A, B*). RGCs, ACs, and their dendritic processes, which are components of the IPL, were localized along the boundaries of these compartments (*Figure 4—figure supplement 2A, B*). Furthermore, RGCs and ACs were observed in the outer part of the retina. At P3 and P5, the interspaces between compartments appeared to be reduced, coinciding with an increase in RGCs and ACs positioned on the inner side of the retina (*Figure 4—figure supplement 2A, B*). By comparison, the positioning of BCs and rod photoreceptors appears to remain disorganized throughout postnatal development (*Figure 4E*). Contrary to our expectation that retinal structure would become increasingly disorganized as development progresses, laminar disruption in the cKO retina appeared more severe at earlier postnatal stages than at later ones, particularly in the IPL, ACs, and RGCs (*Figure 4—figure supplement 2A, B*). Aberrant acellular regions were observed in the cKO retina from P1 through P14 (*Figure 4E*, *Figure 4—figure supplement 2A-C*). Some of these regions contained AC processes and were

surrounded by AC and RGC somata from P3 onward (*Figure 4—figure supplement 2A, B*). Correspondingly, clear signals of rod OS markers were observed in the interior of some acellular regions at P8, P11, and P14 (*Figure 4—figure supplement 2C*). At 1 M, these structures were not observed in the cKO retina (*Figure 2B*). These data suggest that both acellular patch and photoreceptor rosette may be formed in the developing retina, but photoreceptor rosette may degenerate after P14. The OS marker signals were disorganized in the cKO retina at P8, P11, and P14 (*Figure 4—figure supplement 2C*). Moreover, the row of rod and cone photoreceptor synapses was not observed in the cKO retina at P11 (*Figure 4—figure supplement 2D*). These data indicate that proper development of retinal lamination and synapse formation is impaired in the cKO retina.

## Small local field potentials are recorded from the cKO retina

To examine whether the cKO mouse retina can respond to light stimulation, we first recorded electroretinogram (ERG) from anesthetized animals. Under the scotopic condition, a flashlight was applied to eyes in the dark (*Figure 5A left*). Scotopic ERG from the cHet mouse showed a negative a-wave and a following positive b-wave, whereas both waves from the cKO mouse were almost flat. Under the photopic condition, the mouse was adapted to the background light for 10 min, then a flashlight was superimposed on the adapting background (*Figure 5A right*). Photopic ERGs from the cHet mouse showed a small a-wave and a prominent b-wave, whereas those from the cKO mouse were almost flat. These results may be attributed to the scattered distribution of rod and cone photoreceptors, a lack of a uniform photoreceptor layer, and the decrease of OSs and photoreceptor synapses in the cKO retina (see *Figures 2 and 3*).

Then, to examine whether local field potentials can be detected from the cKO retina, we performed multielectrode array (MEA) recordings from the retinas isolated from the cKO and cHet mice. Surprisingly, a flashlight could occasionally evoke micro-ERGs (mERGs) from the cKO retina (SD of the amplitude during 300ms after the flashlight onset; >50 μV, 13/590 electrodes, n=10 retinas from 6 mice; *Figure 5B right*). However, the amplitude of mERGs from the cKO retina was smaller than that from the cHet retina (SD of the amplitude during 200ms after the flashlight onset; >1000 μV, 292/413 electrodes, n=7 retinas from 4 mice; *Figure 5B left*). As the a-wave represents hyperpolarization of the photoreceptors and the b-wave arises from the inner retina, predominantly from ON BCs with a contribution from Müller glial cells (*Perlman, 2020*), the first negative and following positive waves of mERGs may correspond to the a- and b-waves, respectively. To confirm this, L-AP4, an agonist of mGluR6, was bath-applied to suppress the response of ON BCs. In the presence of L-AP4, the second positive wave disappeared, whereas the first negative wave was enhanced in the cHet retina and remained in the cKO retina (*Figure 5B*), indicating that the first negative and following positive waves are the a- and b-waves, respectively. We compared the amplitude and implicit time (time-to-peak latency) of the a- and b-waves and found that both a- and b-waves were significantly smaller in amplitude (*Figure 5C*) and longer in implicit time in the cKO retina than those in the cHet retina (*Figure 5E*). The ratio of b-wave amplitude to a-wave amplitude (b/a) was significantly smaller in the cKO retina than in the cHet retina (*Figure 5D*), suggesting that the efficiency of synaptic transmission from photoreceptors to ON BCs may be decreased in the cKO retina. However, the difference in the implicit time between a- and b-waves was similar in the cKO and cHet retinas (*Figure 5F*), suggesting that signals from photoreceptors to ON BCs may be transmitted properly. These results indicate that despite severe disorganization of the cKO outer retina, photoreceptors can still respond to light stimulation and that the synaptic transmission from photoreceptors to ON-BCs remains functional.

## RGCs in the cKO retina evoke firings to light stimulation

Using the MEA system, we recorded spike discharges from RGCs in retinas isolated from the cKO and cHet mice. The frequency of spontaneous firings in the dark was not significantly different between cKO and cHet retinas (cHet; 12.5±19.2 Hz, n=434, cKO; 10.8±14.6 Hz, n=388, p=0.79, Mann-Whitney U test). We examined whether RGCs in the cKO retina respond to light stimulation. The isolated retina was stimulated by a flash seven times every 8 s, and the peristimulus time histogram (PSTH) was calculated for each RGC after spike sorting. RGCs could be classified into 'ON', 'ON-OFF', 'ON-OFF inhibition', 'OFF', and 'None' types (*Figure 6A and B*; see Materials and methods). All types were found in both cKO and cHet retinas, and the composition of RGC types was similar in both retinas (*Figure 6C*). We also investigated whether RGCs received both rod and cone inputs and found

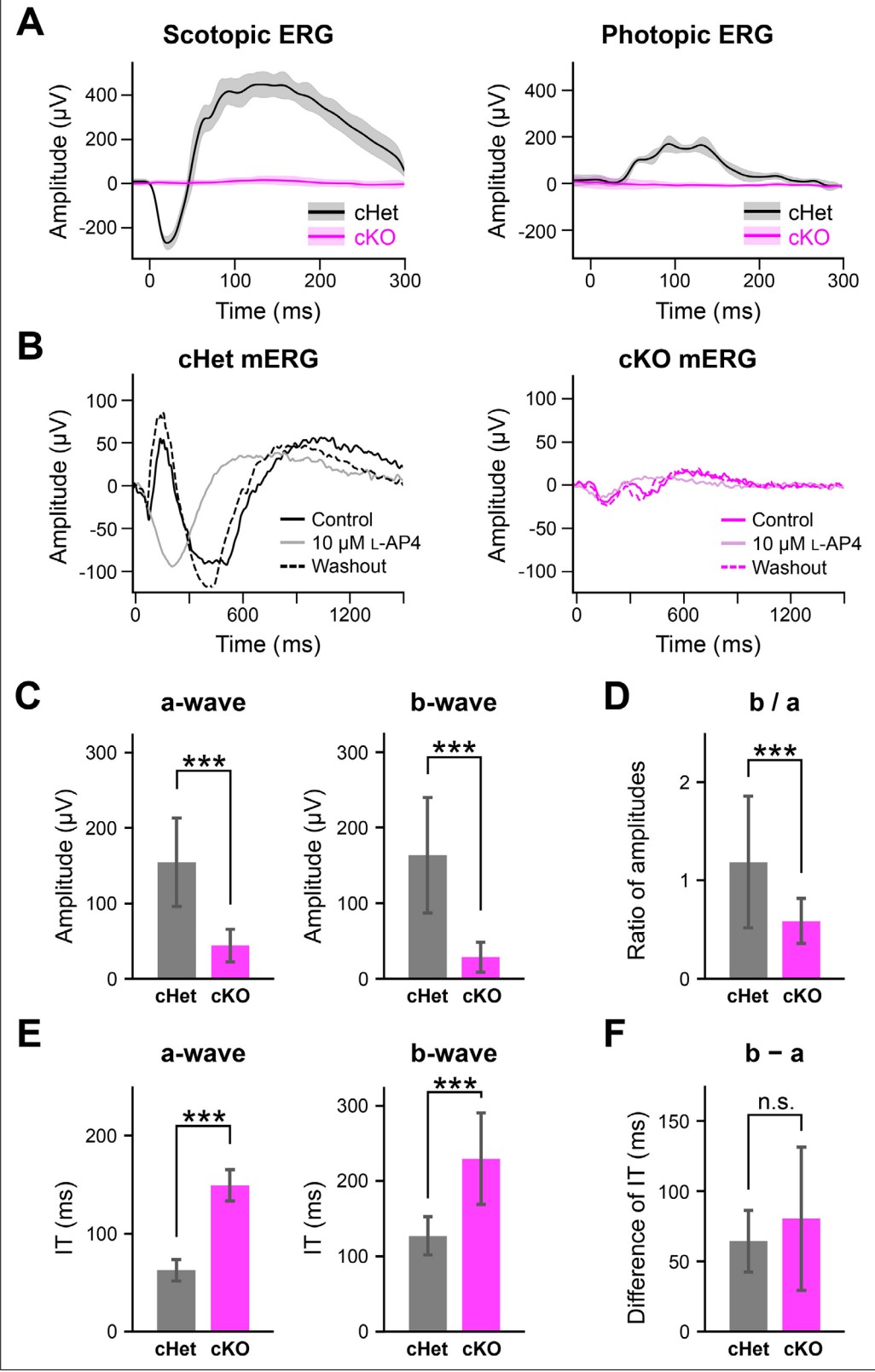

**Figure 5.** Reduction of the a- and b-waves in the cKO retina. (**A**) *Left*, Scotopic ERGs from the anesthetized cHet (n=3; black) and cKO (n=3; magenta) mice. A flashlight (duration; 5ms, white LED, intensity; $1.0\times10^4$ cd/m$^2$) was applied to eyes in the dark three times. Traces: averaged (dark) and SD (pale). *Right*, Photopic ERGs. The anesthetized cHet (black) and cKO (magenta) mice were light (31.6 cd/m$^2$) adapted for 10 min, and then, a

*Figure 5 continued on next page*

*Figure 5 continued*

flashlight (duration; 5ms, intensity; $1.0\times10^4$ cd/m$^2$) was superimposed on the adapted light to eyes 16 times. (**B**) mERGs recorded by MEA from the cHet (black) and cKO (magenta) retinas. A flashlight (duration; 5ms, green LED; $\lambda_{max}$ = 510 nm, intensity; $2.5\times10^4$ photon/s/μm$^2$) was applied to the isolated whole retina three times every 10 s. Control (solid line), L-AP4 (10 μM) (pale), and after washout (dotted line). *Left*, cHet; SD of the amplitude for 200ms after the flash onset; >1000 μV, 292/413 electrodes, n=7 retinas from 4 mice. *Right*, cKO; SD of the amplitude for 300ms after the flash onset; >50 μV, 13/590 electrodes, n=10 retinas from 6 mice. (**C**) *Left*, Amplitude of the a-wave (cHet; 155±58.7 μV, n=292 electrodes, n=7 retinas from 4 mice, cKO; 44.1±21.7 μV, n=13 electrodes, n=10 retinas from 6 mice, Error bars, mean ± SD. ***p<0.001, Mann-Whitney U test). *Right*, Amplitude of the b-wave (cHet; 163±76.5 μV, n=292 electrodes, cKO; 28.4±19.6 μV, n=13 electrodes, p=7.08 x 10$^{-9}$, Mann-Whitney U test). (**D**) Ratio of b-wave amplitude / a-wave amplitude (b / a) (cHet; 1.19±0.67, cKO; 0.59±0.23, Error bars, mean ± SD. ***p<0.001, Mann-Whitney U test). (**E**) *Left*, Implicit time of the a-wave (cHet; 62.6±10.8ms, n=292 electrodes, cKO; 149±16.0ms, n=13 electrodes, p=5.34 x 10$^{-10}$, Mann-Whitney U test). *Right*, Implicit time of the b-wave (cHet; 127±25.3ms, n=292 electrodes, cKO; 229±60.7ms, n=13 electrodes, Error bars, mean ± SD. ***p<0.001, Mann-Whitney U test). (**F**) Difference between implicit time of the b-wave and that of the a-wave (cHet; 64.4±22.0ms, n=292 electrodes, cKO; 80.4±50.9ms, n=13 electrodes, Error bars, mean ± SD. p=0.40, Mann-Whitney U test).

that >60% of RGCs among successfully spike sorted RGCs in both the cKO and cHet retinas received both inputs (*Figure 6D*).

## Receptive fields are characterized in some RGCs

Using pseudorandom checkerboard pattern stimulation, we examined the properties of spatiotemporal RF of RGCs by calculating the spike-triggered average. We could identify RFs in the cKO retina. To check the morphology of the retina where RFs were identified, the retina was removed from the MEA after recording, and sliced retinal sections were immunostained with anti-PKARIIβ, anti-PSD95, and anti-PKCα antibodies (*Figure 7A*). The sectioned region clearly showed impairment of the outer retina, where it was difficult to identify or locate photoreceptors. On the other hand, the inner retina was apparently normal, and both the IPL and the proximal part of the INL could be recognized. Even though the outer retina was severely impaired, we could identify RF of RGCs (*Figure 7B*). However, the cells with clear RF were fewer in the cKO retina (69/388 cells, n=4 retinas) than those in the cHet retina (195/434 cells, n=4 retinas). By fitting an ellipse to each RF, we measured the short and long axes and calculated the RF area (*Figure 7C, D and E*). The RF area in the cKO retina was significantly smaller than that in the cHet retina. We tried to find an antagonistic surround of RFs, but it was difficult to detect a clear surround in both cKO and cHet retinas.

The temporal profile of RFs was calculated, and the principal component analysis was performed. Then, using Uniform Manifold Approximation and Projection (UMAP) for clustering, we could recognize five clusters (*Figure 7F*, cKO; n=4 retinas, cHet; n=4 retinas). The temporal profile belonging to each cluster was superimposed separately (*Figure 7G*). Among five clusters, clusters 1 and 2 did not include the temporal profiles calculated from RGCs in the cKO retina. The temporal profiles of cluster 1 and cluster 2 were ON and OFF types, respectively, each of which had a shorter time-to-peak latency than the other temporal profiles obtained from RGCs in both cKO and cHet retinas (*Figure 7H*). These results indicate that the light-evoked responses of RGCs in the cKO retina are not as refined as cHet. Actually, the rising time of transient responses (determined by an upward deflection soon after light stimulation obtained from the cumulative curve of each PSTH) was significantly longer in the cKO retina (58.5±19.8ms, n=17) than in the cHet retina (43.0±17.8ms, n=23; ***p<0.001, Student's test).

It has been reported that degenerated retinas, such as the *rd1* retinas (*Poria and Dhingra, 2015*), show oscillatory spontaneous firings in RGCs. Thus, we checked whether RGCs in the cKO retina show RGC oscillations. RGCs in the cKO retina showed oscillations (P30; 1.55% of 382 RGCs, 14.0±2.4 Hz, n=4 retinas, P40; 15.0% of 333 RGCs, 9.2±3.8 Hz, n=4 retinas). On the other hand, RCGs in the cHet retina did not show obvious RGC oscillations (0% of 432 RGCs, n=4 retinas).

## *Afadin* cKO mice retain partial visual function

As the MEA analyses suggested partial preservation of retinal functions in the cKO retina, we further examined whether cKO mice retained visual function, using visual behavior tests. We first conducted the optomotor response (OMR) test, a robust method for assessing contrast-based visual function and estimating visual acuity in mice (*Thaung et al., 2002*; *Ye et al., 2021*). The OMR test measures

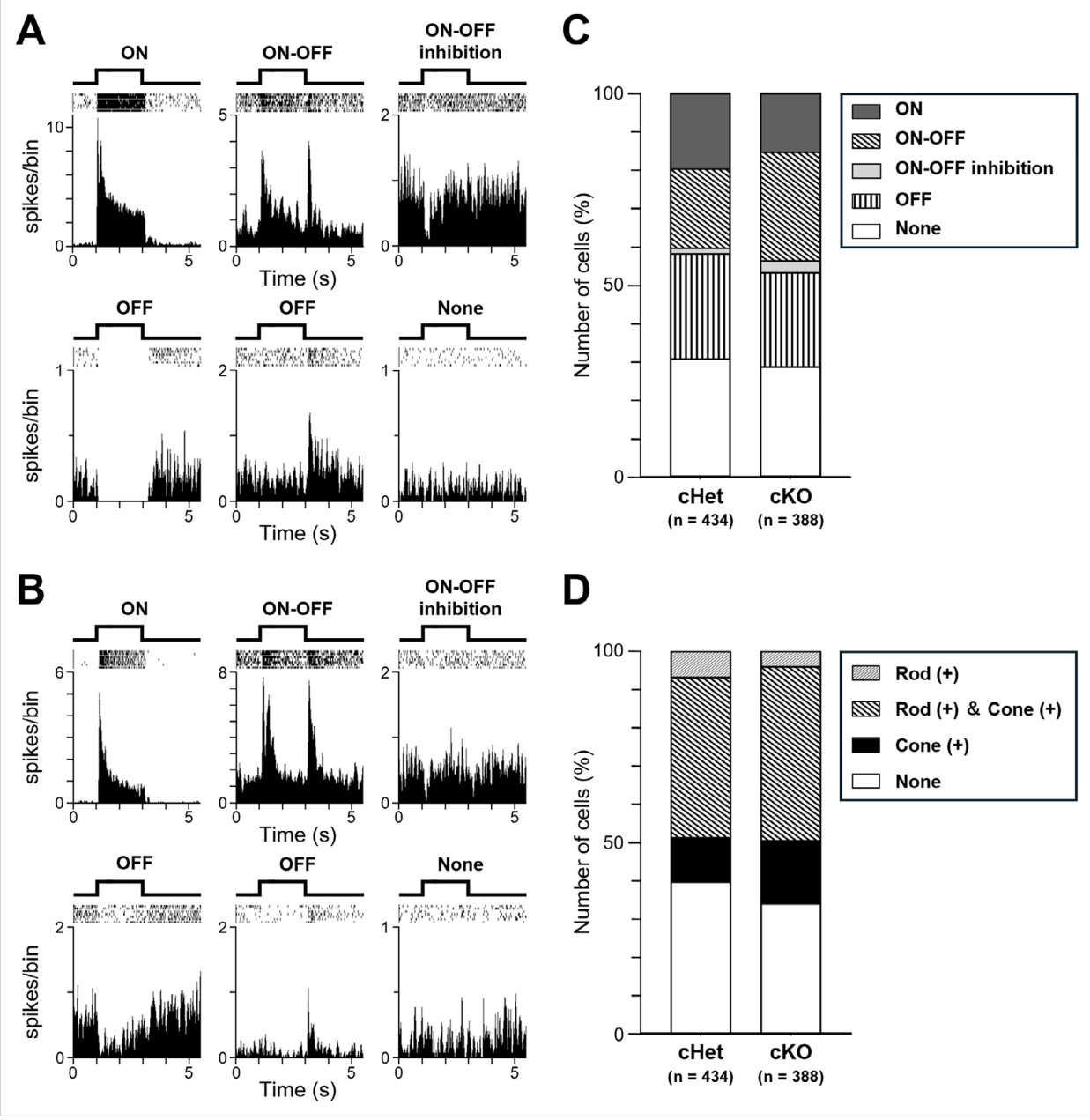

**Figure 6.** RGC classification based on the light-evoked responses. (**A, B**) Representative light-evoked responses. (**A**) The cHet retina. (**B**) The cKO retina. A flash (duration; 2 s, intensity; 24.2 or 27.3 cd/m²) was applied seven times every 8 s. After spike sorting, the raster plots (*top*) and the PSTHs (*bottom*) were created (20ms/bin). Based on the PSTHs, RGC responses were classified into 'ON', 'ON-OFF', 'ON-OFF inhibition', 'OFF', and 'None' types. (**C**) Ratio of responded to non-responded types (cHet; n=434, cKO; n=388). (**D**) Rod and cone inputs to RGCs (cHet; n=434, cKO; n=388). 2 s green LED flash ($\lambda_{max}$ = 510 nm, intensity; 4.8x10³ photons/s/μm²) and UV LED flash ($\lambda_{max}$ = 360 nm, intensity; 3.3x10⁴ photons/s/μm²) were applied 7 times every 8 s to stimulate mainly rods and S cones, respectively.

head-tracking movements in response to a vertical square wave grating stimulus (*Figure 8A*). Mice capable of detecting contrasting motion passing through their visual field exhibit head-tracking movements in response to stimuli (*Kretschmer et al., 2015*; *Mitchiner et al., 1976*; *Schmucker et al., 2005*). Apparent head-tracking movements were observed in the cHet and cKO mice during presentation of drifting grating stimuli at spatial frequencies of 0.056 and 0.1 cycles/degree, but not in those with optic nerve crush (ONC), in which the optic nerve had been mechanically crushed, resulting in vision loss (*Figure 8B and C*). The number of head-tracking movements in response to these stimuli was significantly higher in the cHet and cKO mice than in the ONC mice. At 0.27 cycles/degree, the

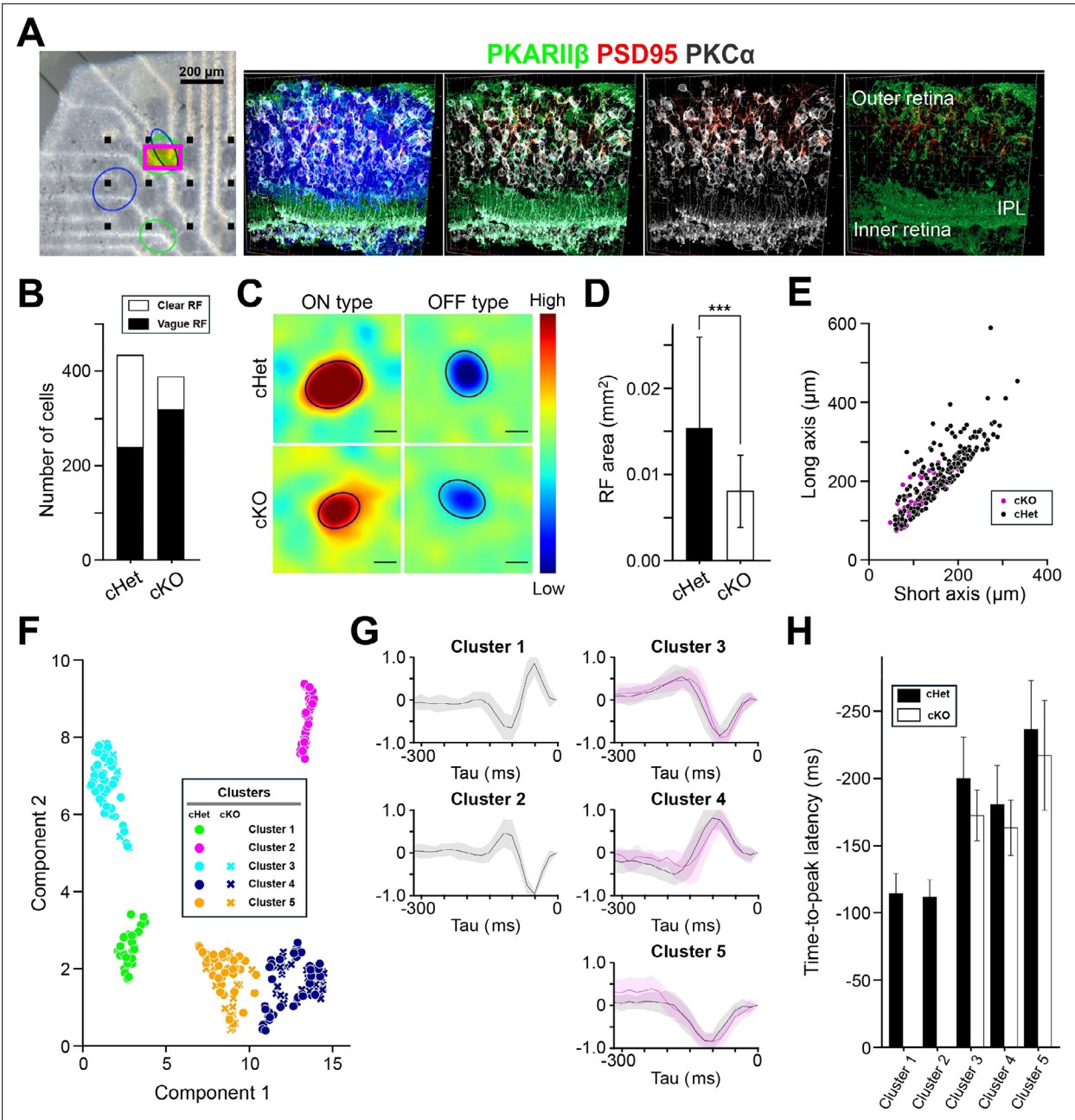

**Figure 7.** Spatiotemporal properties of the RF of RGCs. (**A**) *Left*, An isolated retina on the MEA. Electrodes (black square), mapped RFs (colored oval), and the region where immunohistochemical examination was performed after recording (magenta square). Scale bar 200 µm. *Right*, immunohistochemical staining with anti-PKARIIβ (green), anti-PSD95 (red), and anti-PKCα (white) antibodies. IPL: the inner plexiform layer. (**B**) Number of cells whose RF was clearly observed (cHet; 195/434 cells, n=4 retinas, cKO; 69/388 cells, n=4 retinas). (**C**) The spatial profile of RFs in the cHet (*Top*) and cKO (*Bottom*) retinas. Color scale illustrates high (red) to low (blue) relative to the average (green). Scale bar 100 µm. (**D**) RF size of the cKO retina was significantly smaller than that of the cHet retina (cHet; $0.0307\pm0.0210$ mm$^2$ n=195, n=4 retinas, cKO; $0.0161\pm0.00839$ mm$^2$ n=69, n=4 retinas, Error bars, mean ± SD. ***p<0.001, Mann-Whitney U test). (**E**) Short and long axes of the oval fitted to each RF. (**F**) Clustering of RGCs based on the temporal profile of RFs. Five clusters were visualized (cHet; circle, n=4 retinas, cKO; x, n=4 retinas). (**G**) Temporal profiles from each cluster were superimposed separately (cHet; black, cKO; magenta, mean; solid line, SD: pale line). (**H**) Time-to-peak latency of the temporal profile (cHet; filled bar, cKO; open bar). Mean ± SD. (cHet; cluster 1; −113±15 ms, n=31, cluster 2; −111±13 ms, n=36, cluster 3; −200±31 ms, n=51, cluster 4; −171±19 ms, n=37, cluster 5; −178±28 ms, n=40, cKO; cluster 3; −163±21 ms, n=10, cluster 4; −238±34 ms, n=36, cluster 5; −217±41 ms, n=23).

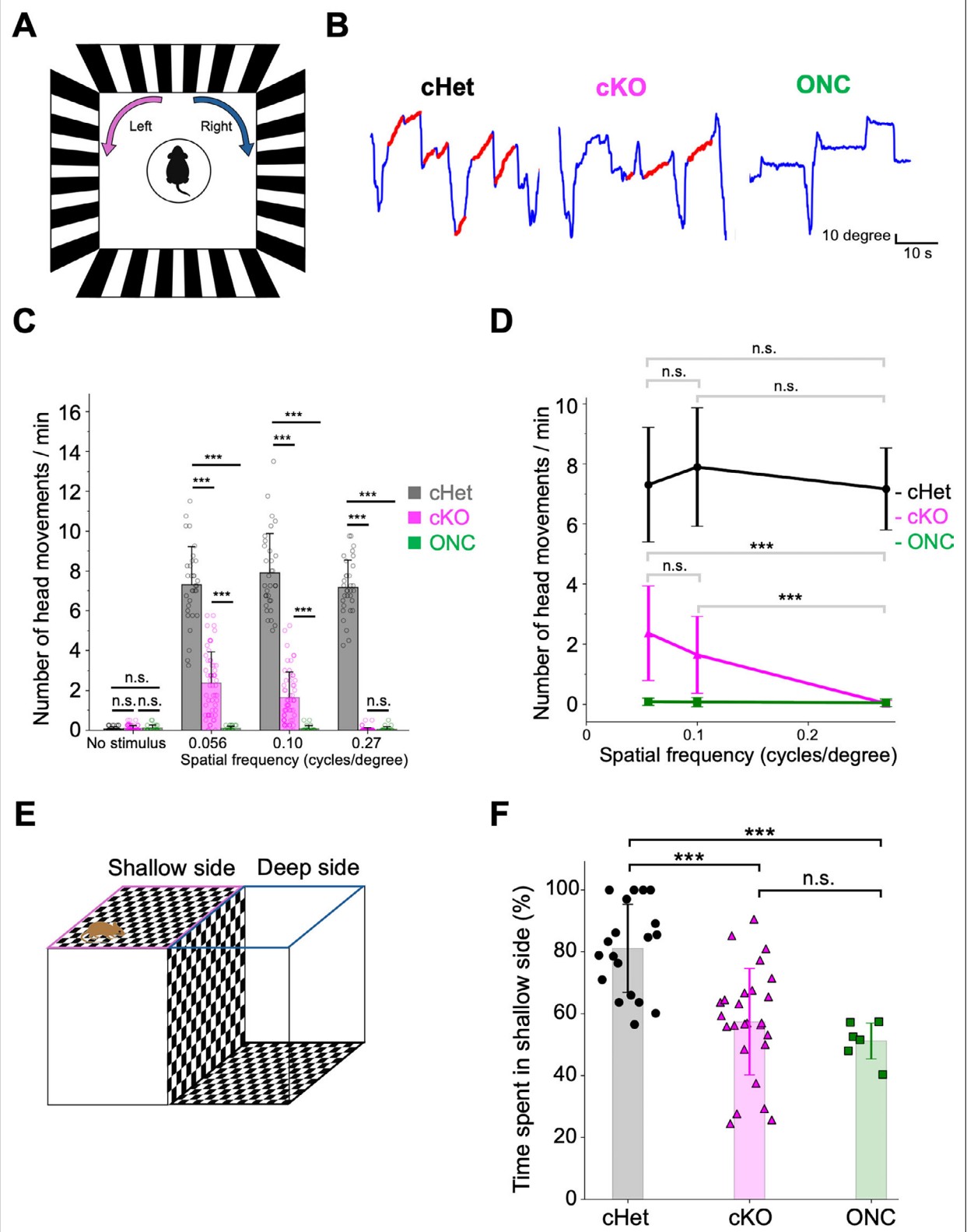

**Figure 8.** Visual function is partially retained in the *Afadin* cKO mice. (**A**) Schematic diagram of the experimental setup for the OMR test. Drifting square-wave gratings were displayed on monitors surrounding the mouse, and head-tracking movements were recorded under photopic conditions. (**B**) Representative traces of head-tracking movements in the cHet, cKO, and ONC mice. Blue lines indicate head movement trajectories, and red segments denote the periods when the head movement aligned with the direction and timing of the visual stimulus. (**C**) Summary of head-tracking movements

*Figure 8 continued on next page*

*Figure 8 continued*

in response to drifting gratings with spatial frequencies of 0.056, 0.10, and 0.27 cycles/degree, as well as under no-stimulus conditions, in the cHet, cKO, and ONC mice (cHet; 0.056 cycles/degree 7.30±1.94, 0.10 cycles/degree 7.89±2.00, 0.27 cycles/degree 7.16±1.39, n=8 mice, cKO; 0.056 cycles/degree 2.36±1.59, 0.10 cycles/degree 1.64±1.29, 0.27 cycles/degree 0.0268±0.103, n=14 mice, ONC; 0.056 cycles/degree 0.0833±0.121, 0.10 cycles/degree 0.0714±0.161, 0.27 cycles/degree 0.0476±0.128, n=7 mice). Error bars, mean ± SD. No stimulus; cHet vs cKO p>0.9999, cHet vs ONC p>0.9999, cKO vs ONC p>0.9999, 0.27 cycles/degree; cKO vs ONC p>0.9999. ***p<0.001, Dunn's test. (**D**) Comparison of head-tracking responses across spatial frequencies (0.056, 0.10, and 0.27 cycles/degree). Statistical comparisons were conducted within each mouse group. No significant differences were detected across spatial frequencies in cHet and ONC mice. In contrast, the cKO mice exhibited a trend toward decreased head-tracking responses with increasing spatial frequency. Error bars, mean ± SD. cHet; 0.056 cycles/degree vs 0.1 cycles/degree p=0.3566, 0.1 cycles/degree vs 0.27 cycles/degree p=0.2606, 0.056 cycles/degree vs 0.27 cycles/degree p>0.9999, cKO; 0.056 cycles/degree vs 0.1 cycles/degree p=0.2577. ***p<0.001, Dunn's test. (**E**) Schematic diagram of the visual cliff test. The apparatus consists of a single transparent platform spanning two arenas with identical checkerboard patterns: one immediately beneath the platform (the shallow side) and the other on the floor (the deep side), creating a visual cliff. Mice are placed at the boundary between the two sides, and their location preference is evaluated. (**F**) Time spent on the shallow side during the visual cliff test. The percentage of time spent on the shallow side was significantly higher in the cHet mice than in the cKO and ONC mice. No significant difference was observed between the cKO and ONC mice (cHet; 81.1±14.6, n=19 mice, cKO; 57.4±17.6, n=26 mice, ONC; 51.1±6.38, n=6 mice). Error bars, mean ± SD. ***p<0.001, one-way ANOVA followed by Tukey's multiple comparisons test.

cKO mice hardly exhibited head-tracking movements, in contrast to the cHet mice, and no significant difference was observed between the cKO and ONC mice under this condition. The number of movements was significantly lower in the cKO mice than in the cHet mice at all three spatial frequencies (*Figure 8C*). There was no significant difference in the number of head-tracking movements across the three spatial frequencies in the cHet mice, whereas in the cKO mice, the number tended to decrease with increasing spatial frequency (*Figure 8D*). These data suggest that the cKO mice can recognize moving visual stimuli, but their ability is inferior to that of the cHet mice.

We further conducted the visual cliff test to evaluate visual performance in a perceptual task. This test exploits rodents' innate tendency to avoid the deep side of the cliff—a plexiglass surface suspended 80 cm above a checkerboard on the floor—and prefer the shallow side, where the plexiglass covered directly atop the checkerboard (*Figure 8E*; *Fox, 1965*). The cHet mice spent approximately 80% of their time on the shallow side, significantly more time than the cKO and ONC mice, while the cKO mice did not differ significantly from the ONC mice in time spent on the shallow side (*Figure 8F*). Nonetheless, some cKO mice exhibited a behavior in which they looked down toward the bottom of the deep side and quickly backed away (*Video 1*). This behavior, observed in the cHet mice but not in the ONC mice, may imply that they can visually recognize the deep side (*Video 2*). Furthermore, the cKO mice, but not the ONC mice, showed a significant preference for the shallow side over the deep side (57.35%, p=0.048, one-sample t-test, figure not shown). These results indicate that visual information from the retina is transmitted to the brain and visual function is partially maintained in *Afadin* cKO mice.

## Ectopic AJs and aberrant localization of progenitor cells are observed in the developing *Afadin* cKO retina

As RGC oscillations were observed in the cKO retina, like the *rd1* retinas with photoreceptor degeneration (*Choi et al., 2014*), we examined whether RGC oscillations were related to cell death in the cKO retina. Using anti-active caspase 3 (an apoptotic cell marker), we counted the number of dead cells immunohistochemically. We found that dead cells were significantly increased in the cKO retina at around P30 (*Figure 9—figure supplement 1A, B*). These findings suggest that the progressive cell death observed in the cKO retina may lead to a further decrease of photoreceptors, resulting in

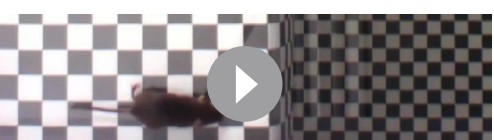

**Video 1.** Behavior of the *Afadin* cKO mouse in the visual cliff test. Excerpt of the cKO mouse behavior during the visual cliff test.

https://elifesciences.org/articles/105627/figures#video1

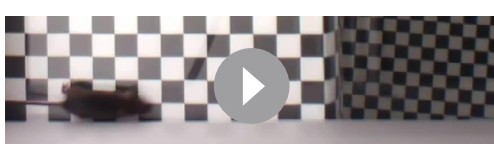

**Video 2.** Behavior of the *Afadin* cHet mouse in the visual cliff test. Excerpt of the cHet mouse behavior during the visual cliff test.

https://elifesciences.org/articles/105627/figures#video2

an increase of oscillatory RGCs at P40. Apoptotic cells were significantly increased in the cKO retina in the developing stage at P1, P3, P5, and P14 (*Figure 9—figure supplement 1B*). In the cKO retina at 1 M, the number of rod photoreceptors and Müller glial cells was significantly reduced to approximately 60% and 70% of that at P14, respectively (P14; rod 98.3±3.8, Müller glial cell 17.2±3.3, n=3, 1 M; rod 57.3±17.3, Müller glial cell 12.4±1.5, n=4, rod p=0.011, Müller glial cell p=0.046 by Student's t-test), in contrast to the cHet retina, in which the number of rod photoreceptors remained unchanged (P14; rod 176.5±14.3, Müller glial cell 17.3±1.4, n=4, 1 M; rod 185.9±17.5, Müller glial cell 15.7±2.3, n=4, rod p=0.43, Müller glial cell p=0.27 by Student's t-test). This reduction may result from ongoing rod cell death even after the completion of differentiation.

Then we investigated the mechanism underlying lamination defects in the cKO retina. We previously demonstrated that the photoreceptor-cell specific *aPKCλ* KO retina shows disruption of AJs and mislocalization of mitotic progenitors in developing stage, followed by severe lamination defects in adults (*Koike et al., 2005*). In the developing retina, AJs are observed in the outer retinal surface, where progenitor cells divide and migrate toward the inner retina. To investigate the possibility that disruption of AJs may lead to abnormal cell migration and lamination, we immunostained the cKO retina at P0 with anti-β-catenin, N-cadherin, nectin-1, and phospho-histone H3S10 (pHH3, a marker of the mitotic cell) (*Figure 9A*). N-cadherin, nectin-1, and β-catenin signals were obviously detected, but these signals were localized in the inner retina with aberrant patterns. These signals were slightly observed in the outer retina. These data indicate that ectopic AJs are present in the cKO retina. Mitotic cells immunostained by pHH3 were localized near the ectopic AJs. The number of pHH3-positive cells was significantly increased to approximately 150% of that in the cHet (*Figure 9B*). At P1, P3, and P5, mitotic cells were also mislocalized to the middle retinal region in the cKO mice, and their distance from the outer retinal surface was significantly larger —approximately 45 μm, 70 μm, and 75 μm, respectively— than that in the cHet (*Figure 9C and D*). Mitotic cells showed a broader distribution in the cKO retina than in the cHet. We further examined the distribution of retinal progenitor cells during development. In the cHet retina, progenitor cells labeled with BrdU were scattered but relatively enriched in the middle region at P1 (*Figure 9E and F*). As the development progressed, the number of progenitor cells decreased, and their distribution was progressively confined to the middle region of the retina. In the cKO retina, progenitor cells were more broadly distributed (*Figure 9E and F*). At P5, no significant difference in the distance was observed between the cHet and cKO retinas. However, progenitor cells in the cKO retina showed a trend toward a broader distribution compared to the cHet retina. Retinal thickness and the number of progenitor cells did not differ significantly between the cHet and cKO retinas at P1, P3, and P5 (*Figure 9—figure supplement 1C, D*). These data suggest that progenitor cells may divide at ectopic regions and migrate aberrantly into the cKO retina, resulting in lamination defects.

## Discussion

In the current study, we showed that the *Afadin* cKO retina exhibits various severe pathological defects, including photoreceptor morphological defects, mislocalization, reduced number of photoreceptors and their synapses, disruption of the outer retinal lamination, ectopic localization of GluR5, aberrant rosette-like structure in the outer retina (*Figures 2–4*). These results suggest that afadin is required for normal retinal lamination, photoreceptor cell morphology, photoreceptor synapse formation, and proper GluR5 localization to OFF BC dendrites. Conditional loss of AJ molecules in the retina has been reported to cause partial or complete disruption of retinal lamination (*Erdmann et al., 2003*; *Fu et al., 2006*; *Masai et al., 2003*). However, the effect of AJ molecule loss on the retinal cell morphology and synapses was unclear. We revealed that loss of afadin significantly but not completely reduces photoreceptor synapses.

In the *Afadin* cKO mouse retina with severe outer retinal disorganization, we detected the a- and b-waves of mERG and the RF of RGCs (*Figures 2, 5 and 7*). These results suggest that some visual information processing may be performed by newly formed and preserved circuits even when the retinal structure is severely impaired and the number of photoreceptors is remarkably decreased. To our knowledge, this is the first report that the RF of RGCs could be formed if several functional photoreceptors connect to BCs, even when the outer retinal lamination is severely disrupted (*Figure 7*). The number and size of RFs observed in the cKO retina were smaller than those in the cHet retina, suggesting that the precise arrangement of retinal cells and lamination is necessary for normal visual

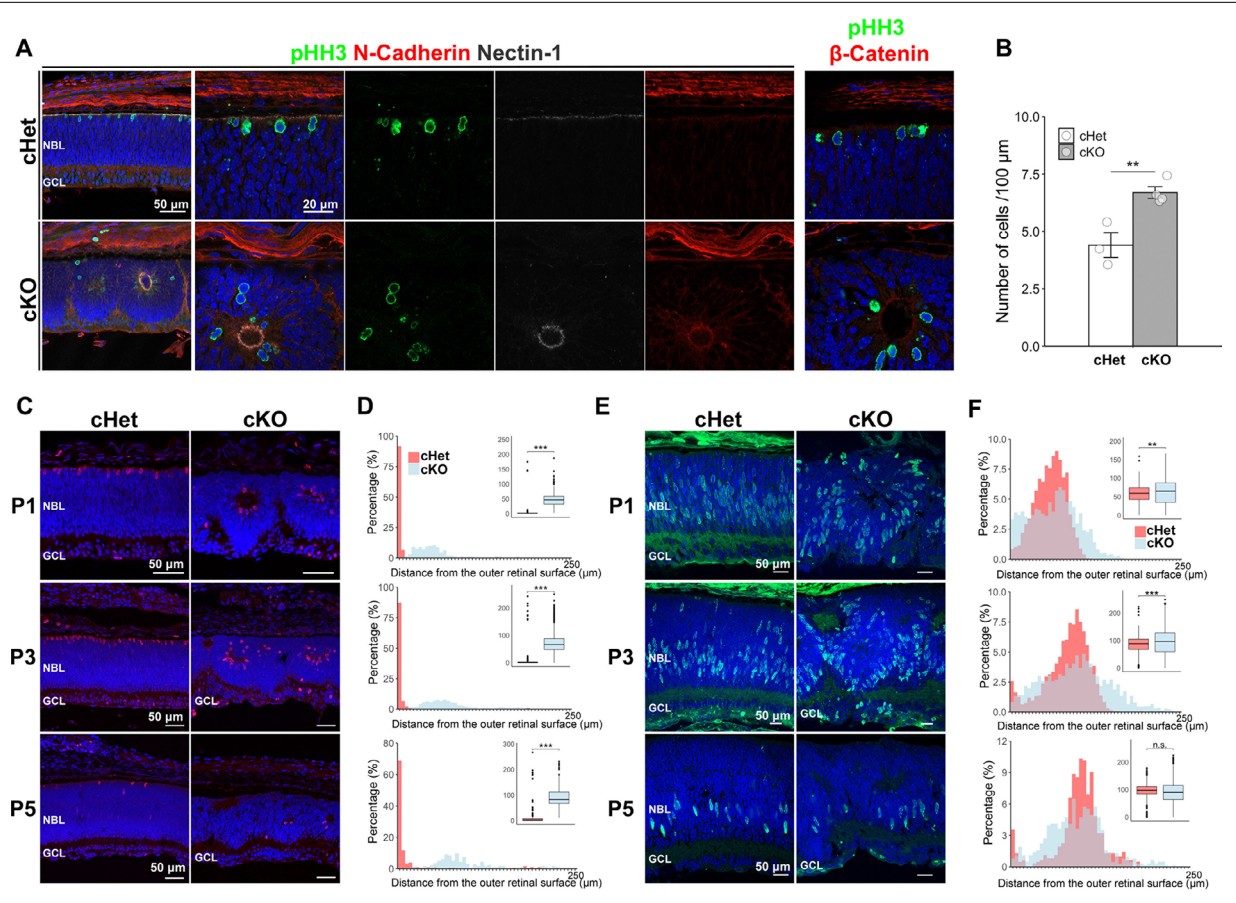

**Figure 9.** Ectopic AJs are observed in the developing *Afadin* cKO retina. (**A**) Immunostaining of the cHet and cKO retinas at postnatal day 0 (P0) with anti-phospho-histone H3S10 (pHH3, mitotic cell marker, green), anti-N-Cadherin (red in left, second left, and second right panels), anti-Nectin-1 (white in left, second left, and third right panels), anti-β-catenin (red in the right panels) antibodies. Ectopic N-cadherin, nectin-1, and β-catenin signals were observed in the cKO retinas, and pHH3-positive cells were localized near these ectopic signals. (**B**) Quantification of pHH3-positive cells per 100 µm width of retinal sections in the cHet and cKO retinas at P0 (cHet; 4.40±0.937, n=3, cKO; 6.69±0.507, n=4). pHH3-positive cells significantly increased in the cKO retina. Error bars, mean ± SD. **p<0.01 by Student's t-test. (**C**) Retinal sections from the cHet and cKO mice immunostained with anti-pHH3 antibody (red) at P1 (upper panels), P3 (middle panels), and P5 (lower panels). (**D**) Distribution of mitotic cells based on their distance from the outer retinal surface in the cHet and cKO mice at P1 (top), P3 (middle), and P5 (bottom). The shortest distance between mitotic cell body and the outer retinal surface was measured using ImageJ. Bin width was 5 µm (0–5 up to 245–250 µm). Inset box plots show the distribution of raw values, with whiskers extending to 1.5 times the interquartile range and dots indicating outliers (cHet; P1 3.76±17.5 µm, n=300 cells from 5 mice, P3 6.73±28.8 µm, n=469 cells from 3 mice, P5 16.7±44.7 µm, n=180 cells from 4 mice, cKO; P1 48.4±24.1 µm, n=326 cells from 4 mice, P3 76.0±41.9 µm, n=902 cells from 4 mice, P5 94.0±44.8 µm, n=163 cells from 4 mice). ***p<0.001 by log1p-transformation followed by a LMM. (**E**) Retinal sections from the cHet and cKO mice immunostained with anti-BrdU antibody (green) at P1 (upper panels), P3 (middle panels), and P5 (lower panels). (**F**) Distribution of progenitor cells based on their distance from the outer retinal surface in the cHet and cKO mice at P1 (top), P3 (middle), and P5 (bottom). Bin width was 5 µm (0–5 up to 245–250 µm). Inset box plots show the distribution of raw values, with whiskers extending to 1.5 times the interquartile range and dots indicating outliers (cHet; P1 58.0±22.1 µm, n=2527 cells from 4 mice, P3 83.5±31.7 µm, n=1904 cells from 4 mice, P5 95.3±32.4 µm, n=620 cells from 5 mice, cKO; P1 62.7±35.2 µm, n=1713 cells from 4 mice, P3 96.5±48.5 µm, n=2025 cells from 3 mice, P5 90.7±40.5 µm, n=630 cells from 3 mice). P5 p=0.49, ***p<0.001, **p<0.01 by log1p-transformation followed by a LMM.

The online version of this article includes the following figure supplement(s) for figure 9:

**Figure supplement 1.** The number of apoptotic cells decreased in the developing cKO retina.

function. It is important to note that light stimulation evoked firing responses in RGCs around the rosette-like structure, and that such RGC firings could not be detected by a large distance between RGC soma and MEA electrode.

We would like to speculate why the RF of RGCs was observed in the *Afadin* cKO retina. First, visual processing may be partially preserved without any remodeling if only a small number of functional photoreceptor-BC-RGC pathways remain intact. It has been shown that the RF of RGCs is detected

even when approximately half of cone photoreceptors are stimulated (*Care et al., 2019*; *Lee et al., 2022*). Thus, in the *Afadin* cKO retina, where cone photoreceptors are reduced to about 40% of the cHet retina, such reduction per se may not critically affect the RF organization. Furthermore, RFs of RGCs are also detected in several mouse models of retinitis pigmentosa, in which rod photoreceptors are degenerated and surviving cone photoreceptors lose their OS discs and pedicles, instead forming abnormal processes resembling synaptic dendrites (*Barhoum et al., 2008*; *Ellis et al., 2023*; *Scalabrino et al., 2022*). A few photoreceptor-BC-RGC pathways (vertical pathways of the retina) are inferred to be maintained in the cKO retina. In some regions, the density of photoreceptors was high enough to make functional synapses between photoreceptor and BC (*Figure 2D*), possibly resulting in the RF of RGCs. Second, synaptic rewiring of interneurons such as BCs may recover retinal function in the *Afadin* cKO retina. Various studies show partial loss of photoreceptors or BCs, or partial silencing of BCs causes rewiring in the developing and mature retinas (*Johnson and Kerschensteiner, 2014*; *Jones et al., 1995*; *Okawa et al., 2014*; *Shen et al., 2020*; *Strettoi et al., 2022*). Also, in the *rd1* retina, neural connections between inner retinal neurons via gap junctions may be enhanced, potentially resulting in the widespread distribution of electrically evoked RGC responses (*Ahn et al., 2022*). These reports suggest that reduced input from photoreceptors and BCs may lead to synaptic reorganization among surviving cells. It is possible that synaptic reorganization triggered by reduced photoreceptor and BC inputs may occur in the *Afadin* cKO retina with severe defects of the photoreceptor synapses and OSs in the developing stages, and that this might contribute to RF formation. Synaptic rewiring may occur between BC dendrites and their closely located photoreceptor terminals in the cKO retina. It may be necessary to count the number of BC dendritic connections to each cone pedicle to verify the synaptic rewiring between photoreceptor and BC, but this was quite difficult because of the remarkably aberrant cone morphology in the cKO retina. As the IPL was relatively intact and BC axon terminals were surrounded by many RGC dendrites, synaptic reconnections between BC axon terminal and RGC dendrite are presumed to be more easily established than those between photoreceptor terminal and BC dendrite in the cKO retina. Thus, several BCs connected with multiple functional photoreceptors may rewire to RGC dendrites and transmit signals efficiently. Our findings imply that very few BCs receiving input from functional photoreceptors responded to light stimulation, and that the output signal can be amplified downstream. The above two possibilities could both contribute to the formation of RGC RFs. Furthermore, we cannot rule out the possibility that gene expression may be changed in the *Afadin* cKO retina, resulting in facilitation of RF formation.

The ratio of responding to non-responding RGCs to light stimulation in the *Afadin* cKO retina was comparable to that in the cHet retina despite notable reduction of photoreceptors (*Figures 4D, 5 and 6*). It seems likely that the number of photoreceptors connected to spreading dendrite tips of one BC may be increased, and that glutamate spillover from BC terminals may contribute to increasing the responsiveness of RGCs in the cKO retina because glutamate transporter EAAT5 and vesicular glutamate transporter vGlut1 were similar in both cKO and cHet retinas (*Figure 4B*, *Figure 3—figure supplement 1D, E*). Glutamate released from BC terminals may spread and bind GluR5, which is ectopically localized in BC processes in the IPL, enhancing the response and expanding the responding area (*Figure 4A and B*). These factors may contribute to the emergence of RF, but the RF size in the cKO retina was still smaller than that in the cHet retina (*Figure 7*). Photoreceptors in the cKO retina were reduced to 30 to 40% of the cHet retina and scattered at 1 M (*Figures 2 and 4D*). The random checkerboard pattern used for STA was projected onto the assumed photoreceptor layer of the normal retina. However, as the morphology and position of photoreceptors were disturbed in the cKO retina (*Figure 2*, *Figure 2—figure supplement 1D*), the remaining photoreceptors would be stimulated by an unfocused image with decreased intensity. Accordingly, photoreceptors possibly responded weakly to unfocused images, and thresholding of noise (increment >+4 SDs, decrement <-2 SDs, determined by the spontaneous firing rate, see Materials and methods) may result in reduction of the RF size. Rod-driven responses were observed in RGCs in the cKO retina (*Figures 5 and 6*). However, it is unlikely that the rod photoreceptors transmit signals to cone photoreceptors through gap junctions in the disrupted cKO outer retina, and thus, rod signals may be transmitted to RGCs via the rod BC-AII amacrine-ON cone BC pathway.

In the cKO retina, the temporal profile of RFs with short time-to-peak latency was not included (Clusters 1 and 2; *Figure 7*), indicating that RGCs in the cKO retina cannot respond to rapid changes in luminance. There are several ways to explain this result. For example, disruption of the outer retinal

lamination may make it difficult to synchronize photoreceptor responses without gap junctions, resulting in dull responses. BCs with fast response kinetics may not be present or not functional in the cKO retina (*Kuo et al., 2024*). Furthermore, glutamate spillover from BC terminals may amplify the signals, but at the same time, the time-to-peak of RGC responses may be delayed. These possibilities should be investigated in the future.

Ectopic continuous AJs formed in the developing *Afadin* cKO retina (*Figure 9A*) are possibly associated with partial disruption of retinal lamination. The $\beta$-catenin cKO retina also shows ectopic continuous AJs in developing stage and the disruption of the outer retinal lamination with relatively intact IPL and GCL structure in adult (*Fu et al., 2006*), suggesting that afadin and β-catenin are required for normal continuous AJ formation at the outer retinal surface but dispensable for maintaining AJs. The photoreceptor-specific *aPKCλ* cKO retina in which AJs are dispersed in the developing stage exhibits disruption of whole retinal lamination despite aPKC $\lambda$ remaining in cells excluding photoreceptors. Individual cells contain AJs, but the AJs are not continuous in the *aPKCλ* cKO retina (*Koike et al., 2005*). These findings, based on the studies using each mutant mouse, imply that continuous AJs are important for the formation of the outer retinal structure. The ONL was severely disrupted in the cKO retina, despite the fact that afadin is not localized in the layer. The disruption is presumed to be caused by defects in AJs in the outer retina during development. In the developing *Afadin* cKO retina, ectopic AJs and mislocalized progenitor cells were observed (*Figure 9*). These defects may lead to abnormal cell migration and positioning, ultimately resulting in ONL disorganization. In contrast, the IPL structure appeared relatively intact in the cKO retina, possibly due to functional redundancy between afadin and other cell adhesion molecules.

In the *Afadin* cKO retina, a significant increase in BCs and a significant decrease in photoreceptors and Müller glial cells were observed at 1 M (*Figure 4D*). The changes in the number and/or marker gene expression of photoreceptors and BCs were also observed in the developing cKO retina (*Figure 4—figure supplement 1C–F*). These results suggest that retinal cell fate was affected in the *Afadin* cKO mice. As retinal cell fate is not altered in the $\beta$-catenin cKO retina (*Fu et al., 2006*), the phenotypic difference between $\beta$-catenin cKO and *Afadin* cKO retinas may imply distinct functional properties of the in AJ molecules.

One of the major concerns of retinal regeneration therapy is that while transplanted photoreceptors can form synapses with donor BCs, the transplanted tissue often shows aberrant structures, such as rosettes (*Yamasaki et al., 2022*). Our study demonstrated that RGCs show RFs and visual behaviors are partially preserved even when the outer retinal lamination is severely disrupted, provided that synaptic connectivity is maintained to a certain extent. Therefore, mice with photoreceptor degeneration that have undergone retinal transplantation might recognize stimulus patterns to some extent, even if aberrant structures are formed after retinal transplantation. Moreover, retinal regeneration therapy could be more effective in restoring vision than previously anticipated. In recent studies, distinct RFs of RGCs were detected in some mouse lines with progressive photoreceptor degeneration or substantial loss of cone photoreceptors or BCs, indicating the robustness of retinal circuitry (*Care et al., 2019*; *Ellis et al., 2023*; *Johnson and Kerschensteiner, 2014*; *Lee et al., 2022*; *Scalabrino et al., 2022*). However, whether visual function is retained in these models remains unclear. Behavioral visual responses are partially preserved in the *Afadin* cKO mice, in which RFs of RGCs are reduced in number and size but clearly detectable. Given that RGCs are intact in the mouse lines used in the aforementioned studies, it is possible that visual functions are preserved to a substantial extent in those models. Although the *Afadin* cKO mice responded to certain visual stimuli, the extent of what visual patterns or stimuli they can perceive remains unclear. Further detailed analyses of visual functions are required in the future.

## Materials and methods

### Key resources table

| Reagent type (species) or resource | Designation | Source or reference | Identifiers | Additional information |
|---|---|---|---|---|
| Strain, strain background (*Mus musculus*) | *Afdn*flox | *Majima et al., 2009* | | |

*Continued on next page*

*Continued*

| Reagent type (species) or resource | Designation | Source or reference | Identifiers | Additional information |
|---|---|---|---|---|
| Strain, strain background (*Mus musculus*) | Dkk3-Cre | *Sato et al., 2007*; *Yamamoto et al., 2020* | RBRC05427 | |
| Strain, strain background (*Mus musculus*) | R26R-H2B-EGFP | *Abe et al., 2011* | CDB0203K | |
| Antibody | anti-α-afadin (mouse monoclonal) | Abcam | Cat# ab90809 RRID:AB_2049761 | IF (1:100), WB(1:1000) |
| Antibody | anti-nectin1 (Rat monoclonal) | MBL | Cat# D146-3, RRID:AB_590847 | IF (1:100) |
| Antibody | anti-nectin2 (Rat monoclonal) | MBL | Cat# D083-3, RRID:AB_590848 | IF (1:100) |
| Antibody | anti-nectin3 (Rat monoclonal) | MBL | Cat# D084-3 RRID:AB_592587 | IF (1:100) |
| Antibody | anti-PKCα (Mouse monoclonal, Rabbit polyclonal) | Sigma-Aldrich | Cat# P5704, RRID:AB_477375 Cat# P4334, RRID:AB_477345 | IF (1:1000, 1:10000) |
| Antibody | anti-SCGN (Goat polyclonal) | R&D Systems | Cat# AF4878, RRID:AB_2269934 | IF (1:2000) |
| Antibody | anti-Arr3 (Rabbit polyclonal) | Millipore | Cat# AB15282, RRID:AB_1163387 | IF (1:1000) |
| Antibody | anti-Calbindin (Rabbit polyclonal) | Millipore | Cat# PC253L-100, RRID:AB_213554 | IF (1:200) |
| Antibody | anti-ChAT (Goat polyclonal) | Millipore | Cat# AB144P, RRID:AB_2079751 | IF (1:50) |
| Antibody | anti-Bassoon (Mouse monoclonal) | Enzo | RRID:AB_1860018 | IF (1:1000) |
| Antibody | anti-mGluR6 (Guinea pig polyclonal) | Current study | | IF (1:3000) |
| Antibody | anti-PSD95 (Mouse monoclonal) | Synaptic Systems | Cat# 124 014, RRID:AB_2619800 | IF (1:3000) |
| Antibody | anti-GluR5 (Goat polyclonal) | Steve H. DeVries | | IF (1:2000) |
| Antibody | anti-PKARIIβ (Mouse monoclonal) | BD Biosciences | Cat# 610625, RRID:AB_397957 | IF (1:500) |
| Antibody | anti-Calretinin (Mouse monoclonal) | Millipore Cat#MAB1568 | RRID:AB_94259 | IF (1:5000) |
| Antibody | anti-EAAT5 (Rabbit polyclonal) | Sigma-Aldrich | Cat# HPA049124 RRID:AB_2680643 | IF (1:100) |
| Antibody | anti-vGlut1 (Guinea pig polyclonal) | Millipore | Cat# AB5905, RRID:AB_2301751 | IF (1:6000) |
| Antibody | anti-HPC-1 (Mouse monoclonal) | Sigma-Aldrich | Cat# S0664, RRID:AB_477483 | IF (1:10,000) |
| Antibody | anti-active caspase-3 (Rabbit polyclonal) | R&D Systems | Cat# AF835, RRID:AB_2243952 | IF (1:300) |
| Antibody | anti-Chx10 (Rabbit polyclonal) | *Hori et al., 2019* | | IF (1:100) |
| Antibody | anti-Otx2 (Goat polyclonal) | R&D Systems | Cat# AF1979, RRID:AB_2157172 | IF (1:200) |
| Antibody | anti-Glutamine synthetase (Mouse monoclonal) | Millipore | Cat# MAB302, RRID:AB_2110656 | IF (1:500) |

*Continued on next page*

*Continued*

| Reagent type (species) or resource | Designation | Source or reference | Identifiers | Additional information |
|---|---|---|---|---|
| Antibody | anti-CD31 (Rat monoclonal) | BD Biosciences | Cat# 557355, RRID:AB_396660 | IF (1:100) |
| Antibody | anti-Lhx2 (Goat polyclonal) | Santa Cruz Biotechnology | Cat# sc-19344, RRID:AB_2135660 | IF (1:100) |
| Antibody | anti-RBPMS (Rabbit polyclonal) | GeneTex | Cat# GTX118619, RRID:AB_10720427 | IF (1:500) |
| Antibody | anti-Ki67 (Mouse monoclonal) | BD Biosciences | Cat# 556003, RRID:AB_396287 | IF (1:100) |
| Antibody | anti-Tuj1 (Mouse monoclonal) | BioLegend | Cat# 801201, RRID:AB_2313773 | IF (1:500) |
| Antibody | anti-AP2α (Mouse monoclonal) | DSHB | Clone 3B5-c RRID:AB_528084 | IF (1:1000) |
| Antibody | anti-Rom1 (Rabbit polyclonal) | Robert Molday | | IF (1:10) |
| Antibody | anti-Rhodopsin (Rabbit polyclonal) | St John's Laboratory | Cat# STJ95452 | IF (1:100) |
| Antibody | anti-S-opsin (Goat polyclonal) | Santa Cruz Biotechnology | Cat# sc-14363, RRID:AB_2158332 | IF (1:500) |
| Antibody | anti-M-opsin (Rabbit polyclonal) | Millipore | Cat# AB5405, RRID:AB_177456 | IF (1:500) |
| Antibody | anti-GFP (Chicken polyclonal) | Aves Labs | Cat# GFP-1010, RRID:AB_2307313 | IF (1:1000) |
| Antibody | anti-β-catenin (Mouse monoclonal) | BD Biosciences | Cat# 610153, RRID:AB_397554 | IF (1:1000) |
| Antibody | anti-N-cadherin (Mouse monoclonal) | BD Biosciences | Cat# 610920, RRID:AB_2077527 | IF (1:500) |
| Antibody | anti-BrdU (Mouse monoclonal) | BD Biosciences | Cat# 347580, RRID:AB_400326 | IF (1:100) |
| Antibody | anti-phospho-histone H3 (Rabbit polyclonal) | Millipore | Cat# 06–570, RRID:AB_310177 | IF (1:2000) |
| Software, algorithm | MATLAB (MathWorks) | MathWorks | RRID:SCR_001622 | |
| Software, algorithm | Psychtoolbox-3 | http://psychtoolbox.org/ | RRID:SCR_002881 | |
| Software, algorithm | Python3 | https://www.python.org/ | RRID:SCR_008394 | |

## Animals

All animal experimental protocols were conducted in accordance with local guidelines and the ARVO statement on the use of animals in ophthalmic and vision research. These procedures were approved by the Institutional Safety Committee on Recombinant DNA Experiments (approval ID R4016) and Animal Experimental Committees (approval ID BKC2022-017) of Ritsumeikan University. Mice were kept in the temperature-controlled room at 25°C with a 12 hr/12 hr day/night cycle. Freshwater and rodent diets were always provided.

WT, *Afadin* cHet, and cKO mice on 129S6/SvEvTac background, and R26R-H2B-EGFP mice on C57BL/6 J background were used in this study. The *Afdn^flox^* mice (**Majima et al., 2009**), *Dkk3-Cre* mice (**Sato et al., 2007**; **Yamamoto et al., 2020**), and R26R-H2B-EGFP mice (**Abe et al., 2011**,CDB Accession number. CDB0203K), have been described previously. Exon 2 of the *Afadin* gene is flanked by loxP sites in the *Afdn^flox^* mice, and Cre activity is detected in the retina of the *Dkk3-Cre* mice from embryonic day 10.5 (E10.5). In R26R-H2B-EGFP mice, Cre-dependent, nuclear-localized green fluorescent protein targeted to the Rosa26 locus. Mice of either sex were used for all animal experiments. The stages used for individual experiments are described in figure legends.

## Immunohistochemistry

Immunohistochemical analysis was performed as described previously (*Hori et al., 2019*; *Kubo et al., 2021*). In brief, the retina used for MEA recordings and isolated mice eyes was fixed with 4% PFA in PBS for 30 min at room temperature. After three-time washes, retinas were cryoprotected by 30% sucrose in PBS overnight, embedded in an OCT compound (Sakura, Japan), frozen on dry ice, and sectioned at 20 µm of thickness. Whole-mount immunostaining was performed as previously described with some modifications (*Ueno et al., 2018*). The retinas were gently peeled off from the sclera, fixed with 4% PFA in PBS for 30 min at room temperature, and washed three times. Retinal sections and whole retinas were soaked in blocking buffer (5% NDS, 0.1% Triton X-100, in 1x PBS) for 1–2 hr at room temperature and incubated with primary antibodies in blocking buffer 1 or 2 overnight at 4°C. The sections were washed with PBS three times and incubated with fluorescent dye-conjugated secondary antibodies and DAPI (1:1000) for more than 2 hr at room temperature or overnight at 4°C under the light-shielded condition. The specimens were observed using a laser confocal microscope (LSM900; Carl Zeiss, Germany). For BrdU staining, mice were given an intraperitoneal injection of 20 mg/kg BrdU and sacrificed 2 hr later. Retinal sections were then pretreated with 2 N HCl at 37°C for 30 min before blocking. The antibodies and dilution ratios were as follows: anti-l-afadin (ab90809, Abcam, UK, 1:100), anti-nectin1 (D146-3, MBL, Japan, 1:100), anti-nectin2 (D083-3, MBL, Japan, 1:100), anti-nectin3 (D084-3, MBL, Japan, 1:100), anti-PKCα (P5704, Sigma, USA, 1:1000, P4334, Sigma, 1:10,000), anti-SCGN (AF4878, R&D systems, USA, 1:2000), anti-Arr3 (AB15282, Millipore, USA, 1:1000), anti-Calbindin (PC253L-100, Millipore, USA, 1:200), anti-ChAT (AB144P, Millipore, USA, 1:50), anti-Bassoon (SAP7F407, Enzo, USA, 1:1000), anti-mGluR6 (current study, 1:3000), anti-PSD95 (#124 014, Synaptic Systems, Germany, 1:3000), anti-GluR5 (Grik1, gift from Steve H DeVries, 1:2000), anti-PKARIIβ (#610625, BD biosciences, USA, 1:500), anti-Calretinin (PC235L-100UCN, Millipore, USA, 1:5000), anti-EAAT5 (HPA049124, Sigma, USA, 1:100), anti-vGlut1 (AB5905, Millipore, USA, 1:6000), anti-HPC-1 (S0664, Sigma, USA, 1:10000), anti-active caspase3 (AF835, R&D Systems, USA, 1:300), anti-Chx10 (*Hori et al., 2019*, 1:100), anti-Otx2 (AF1979, R&D systems, USA, 1:200), anti-Glutamine synthetase (GS, MAB302, Millipore, USA, 1:500), anti-CD31 (#557355, BD Biosciences, USA, 1:100), anti-Lhx2 (sc-19344, Santa Cruz Biotechnology, USA, 1:100), anti-RBPMS (GTX118619, GeneTex, USA, 1:500), anti-Ki67 (#556003, BD Biosciences, USA, 1:100), anti-Tuj1 (#801201, BioLgend, USA, 1:500), anti-AP2α (3B5-c, DSHB, USA, 1:1000), anti-Rom1 (gift from Robert Moldey, 1:10), anti-Rhodopsin (STJ95452, ST John's Laboratory, UK, 1:100), anti-S-opsin (sc-14363, Santa Cruz Biotechnology, USA, 1:500), anti-M-opsin (AB5405, Sigma, USA, 1:500), anti-GFP (GFP-1010, Aves labs, USA, 1:1000), anti-β-catenin (#610153, BD Biosciences, USA, 1:1000), anti-N-cadherin (#610920, BD Transduction Laboratories, USA, 1:500), anti-BrdU (#347580, BD Biosciences, USA, 1:100), and anti-phospho-histone H3S10 (#06–570, Millipore, USA, 1:2000) antibodies.

## Counting of retinal cells and synapses

The cell number of each retinal cell type was counted around the area 500 µm away from the optic nerve using immunostained retinal sections, and we calculated the cell number per 100 µm width of retinal section. Each retinal cell type was identified as follows: rod (Otx2 signal at the nuclear periphery), cone (Arr3+, Otx2 signal in the soma), BC (Chx10+), horizontal cell (Calbindin + and AP2α-), AC (AP2α+), RGC (RBPMS+), and Müller glial cell (Lhx2+).

It was difficult to count the number of synapses between photoreceptor and BC in 2D images of the cKO retina because the direction of the photoreceptor synapse was aberrant. Thus, the number of synapses was counted using 3D images reconstituted from immunostained vertical sections. The image of vertical sections was acquired with a width (x-axis) of 126.8 µm, a length (y-axis) of 126.8 µm, and a thickness (z-axis) of 12 µm. Then, the number of synapses per 1 mm$^2$ of retinal surface was calculated. The regions distal to the IPL were analyzed because photoreceptor synapses were not observed below the IPL. Since the synapses between the rod photoreceptor and BC in the cHet retina were highly dense, we counted them every 1.8 µm using 2D images and integrated them. We confirmed that the number of synapses was not different between 2D and 3D images.

## Processing of tissues for electron microscopy

The eyeball was removed, cut in the cornea, and fixed with 2.5% glutaraldehyde in PBS for 2.5 hr on ice, and with 2% osmium tetroxide for 2 h. Retinas were washed with 1 x PBS, and dehydrated with

a graded series of ethanol, followed by propylene oxide, and embedded in Epon 812. Seventy-nm ultrathin sections were cut by an ultramicrotome (Ultracut E; Reichert-Jung, Germany), mounted on nickel grids, and stained with 2% uranyl acetate for 4 hr and with nitrate for 5 min. Retinal sections were observed by transmission electron microscope (H-7500; HITACHI Co, Japan).

## Reverse transcription real-time quantitative PCR (RT-qPCR)

Total retinal RNA was isolated using Isogen II reagent (Nippon Gene, Japan) following the manufacturer's instructions, and reverse transcribed into cDNA using SuperScript II Reverse Transcriptase (Thermo Fisher Scientific, USA) with random hexamers and oligo (dT) 12–18 primers. Diluted cDNA was used as a template for qPCR using a TB Green Premix Ex Taq II (Tli RNaseH Plus) (Takara Bio, Japan) according to the manufacturer's instructions, on Thermal Cycler Dice Real Time System II (Takara Bio, Japan). The primer sequences are listed in *Supplementary file 1*.

## Western blot analysis

Retinal tissues were lysed in RIPA buffer (50 mM Tris-HCl pH7.6, 150 mM NaCl, 1 mM EDTA, pH8.0, 1% Nonidet-P40, 1% Sodium Deoxycholate, 0.1% SDS), cooled for 10 min on ice, and centrifuged at 14,000 rpm for 10 min at 4°C. Then, a 3 x sample buffer was added to the supernatant, and it was boiled for 10 min at 95°C. Samples were separated by SDS-PAGE and transferred to PVDF membranes (Millipore, USA) using a wet transfer cell (Bio-Rad, USA). The membranes were soaked in blocking buffer (5% skim milk (W/V) and 0.1% Tween 20 in TBS) and incubated with anti-l-afadin antibody (1:1000) overnight at 4°C or horseradish peroxidase-conjugated anti-GAPDH antibody (1:5000) for 2 hr at room temperature. After washing with TBS/0.1% Tween-20 four times for 10 min each, the membranes were incubated with horseradish peroxidase-conjugated anti-mouse IgG antibody for 2 hr at room temperature. Signals were detected using ImmunoStar LD (Fujifilm Wako, Japan).

## Antibody production

cDNA fragments encoding a C-terminal portion of mouse mGluR6 (853–871 residues) were subcloned into *pGEX4T-1* plasmid (Amersham, UK). The GST-mGluR6 fusion protein was expressed in *Escherichia coli* strain BL21 and purified with glutathione Sepharose 4B (GE Healthcare, USA) according to the manufacturer's instructions. The anti-mGluR6 antibody was obtained by immunizing guinea pigs with the purified GST-mGluR6.

## ERG recordings

Mice were dark-adapted for more than 1 hr and then anesthetized deeply by intraperitoneally injecting mixed anesthesia consisting of (in mg/kg) 0.3 medetomidine, 4 midazolam, and 5 butorphanol tartrate. We also administered 0.1% tropicamide and 0.1% phenylephrine to the eyes to dilate the pupil. ERGs were recorded via contact lens electrodes (Mayo Corporation, Japan), and data were sampled at 1250 Hz by using the PuREC system (Mayo Corporation, Japan). Reference and ground electrodes were placed in the mouth and on the tail, respectively. Full-field stimulation was presented via an optical stimulator equipped with white LED (RMG; Mayo Corporation, Japan), and its light intensity and duration were controlled by an LED visual stimulator (LS-100; Mayo Corporation, Japan). Scotopic ERGs were recorded by applying 5 ms flash ($1.0 \times 10^4$ cd/m$^2$) three times in the dark. After 10 min light adaptation (31.6 cd/m$^2$), photopic ERGs were recorded by superimposing 5 ms flash ($1.0 \times 10^4$ cd/m$^2$) sixteen times on the adapting background. After noise reduction by a program installed in the PuREC system, ERGs were low-pass filtered (<50 Hz) on Python 3 and then analyzed.

## MEA recordings

The retinal preparation was made as described previously (*Takeuchi et al., 2018*). In brief, a dark-adapted (>1 hr) mouse was sacrificed by cervical dislocation under a dim red light, and its eyes were enucleated. Under a stereomicroscope equipped with an infrared (IR) image converter (V6833P, Hamamatsu Photonics, Japan) and IR illumination (HVL-IRM, Sony, Japan), the retina was isolated from the eye, and the dorsal retina was placed ganglion cell layer side down onto the MEA (Multichannel Systems, Germany, 60pMEA200/30iR-Ti: 60 electrodes, electrode size 30x30 µm, inter-electrode distance 200 µm) and attached to the MEA by suction using a vacuum pump (Constant Vacuum Pump, Multichannel Systems, Germany). When another type of MEA (Multi Channel Systems, Germany,

60MEA200/30iR-Ti-gr: 60 electrodes, electrode size 30×30 µm, inter-electrode distance 200 µm) was used, a piece of anodisc (13 mm, 0.2 µm, 6809–7023, Whatman, UK) and a weight with nylon fibers were placed on the retina to improve the contact with MEA. Then, the MEA chamber was constantly superfused with Ames' medium (A1372-25, United States Biological, USA) bubbled with 95% $O_2$/5% $CO_2$ at the rate of 6.0 mL/min at 32°C. After >30 min superfusion, we started recordings. Signals were amplified, sampled at 20 kHz, and stored using MEA2100-Lite-System (Multichannel Systems, Germany). In a pharmacological experiment, L-(+)–2-amino-4-phosphonobutyric acid (10 µM; L-AP4, 23052-81-5, Tocris, UK) dissolved in Ames' medium was bath-applied to the retina.

Light stimulus generated by Psychtoolbox-3 on MATLAB was presented on a monitor display (P2314H, Dell, Japan), and projected through optics on the photoreceptor layer. Flashes (24.2 or 27.3 cd/m$^2$, 2 s in duration, 1600 µm in diameter) were presented seven times every 8 s. In some experiments, the retina was stimulated by 5 ms diffuse green LED light (2.5x10$^4$ photons/s/µm$^2$ for mERG recordings, 4.8x10$^3$ photons/s/µm$^2$ for rod stimulation: LED; $\lambda_{max}$ = 518 nm, IF filter; 510 nm, FWHM 10 nm, #65–697, ND filter; OD 1.0 #47–207, Edmund Optics, USA) or 5 ms diffuse UV LED light (3.3x10$^4$ photons/s/µm$^2$ for cone stimulation; LED; $\lambda_{max}$ = 370 nm, KED365UH, IF filter; 360 nm, FWHM 10 nm #67–827, Edmund Optics, USA) controlled by a function generator (WF1973, NF Corporation, Japan) three times every >10 s.

## Spike sorting

For spike sorting, the algorithm described on Python3 was used. The data were high pass-filtered (>300 Hz), and minimum values were detected by using scipy.signal.argrelmin function. To decide whether each detected minimum value reflects the peak of spike event, we used a threshold value defined by the following formula (*Quiroga et al., 2004*).

$$threshold = -4 \times median\,(|x|\,/0.6745),$$

where *x* represents each data point, *as* our data were recorded extracellularly, the threshold polarity was reversed. The detected values (>threshold) were judged as spike events, and the waveform of spikes was obtained from the data points between –1 and + 2ms from each peak. The 1st and 2nd differences (numpy.diff) of each waveform were joined, and then principal component analysis (PCA) was carried out for feature compression to two dimensions. Then, clustering was performed using the template matching method (*Zhang et al., 2004*) and UMAP. To check the accuracy of spike sorting, we performed autocorrelation analysis to confirm the presence of a refractory period (the number of spike events at ± 1ms bin was less than 1% of total spike events).

## Classification of RGCs based on PSTH

Using the MEA system, we recorded firing responses of RGCs to light stimulation (0.24 or 0.27 cd/m$^2$ in intensity, 2 s in duration) presented 7 times from the monitor and calculated the PSTHs with 20 ms bin. The PSTH was smoothed by a Gaussian filter (kernel size: 3 bins, σ=1). The light-evoked response was defined when the firing rate of the PSTH exceeded the threshold (increment >+4 SDs, decrement <-2 SDs) determined by the spontaneous firing rate for 2 s before light stimulation. RGCs were classified into 'ON' (increment within 2 s after light onset), 'ON-OFF' (increment within 2 s after light onset and offset), 'ON-OFF inhibition' (decrement within 2 s after light onset and offset), 'OFF' (decrement within 2 s after light onset and/or increment within 2 s after light offset), and 'None' (no clear change in firing rate) types.

To determine the type of photoreceptor inputs, a 2 s flash of green LED ($\lambda_{max}$ = 510 nm, 4.8x10$^3$ photons/s/µm$^2$) and 2 s flash of UV LED ($\lambda_{max}$ = 360 nm, 3.3x10$^4$ photons/s/µm$^2$) were applied seven times at 8 s interval serially. We classified RGCs into 'Rod (+)' (responded to green LED), 'Cone (+)-Rod (+)' (responded to both green and UV LEDs), 'Cone (+)' (responded to UV LED), and 'None' responsive types.

## Receptive Field

The reverse correlation method was used to detect the receptive field (RF). Pseudorandom checkerboard patterns (32×32 pixels; black/white = 1; pixel size 50×50 µm) were applied to the retina at 60 Hz. The checkerboard images that preceded each spike event (20 frames) were averaged (spike-triggered average: STA). The intensity of each pixel was +1 or 0, and thus the area with pixels that

exceeded + 5 SDs of the noise intensity (~0.5: the SD calculated from the frame at t=0) was judged to be an RF, and the pixel with maximal intensity was defined as the RF center. An ellipse was fitted to the area, and its long and short axes, as well as the area, were calculated. To estimate the temporal profile of RF, the mean intensity of 5×5 pixels in the RF central region was obtained from a series of averaged frames. The temporal profile of RF was normalized, assuming that the mean of the frame at t=0 was 0. The normalized temporal profiles were compressed into two dimensions by using UMAP, and then k-means clustering was performed.

## Visual behavioral tests

We examined the visual behavior of the *Afadin* cHet, *Afadin* cKO, and ONC (optic nerve crush) mice. The optic nerve crush was performed as previously described (*Cameron et al., 2020*), and successful nerve damage was confirmed by absence of a pupillary light reflex.

The optomotor response (OMR) test was performed as described previously with slight modifications (*Kretschmer et al., 2015*; *Kretschmer et al., 2013*; *Warwick et al., 2024*). Mice were placed on a central platform surrounded on all four sides by monitors, with a mirror positioned on the floor beneath the platform. Stimuli were comprised of vertical square wave gratings (white; 116.2 cd/m$^2$, black; 1.66 cd/m$^2$, Michelson contrast; 97.2) with spatial frequency (0.056, 0.10, or 0.27 cycles/degree) manipulated to induce the illusion of a virtual cylinder with identical spatial frequency across the entire display at a constant speed of 12 degree/second. Each trial consisted of 2 min of drifting gratings, followed by either 30 s or 2 min of a uniform white screen (88.6 cd/m$^2$), and then another 2 min of drifting gratings in the opposite direction. This trial was repeated three or four times. Head movements were tracked using a camera (29.97 fps, AN-S093, KEIYO, Japan) and the number of head movements consistent with the stimulus direction and speed was counted in a blind manner.

For the visual cliff test, mice were placed along the centerline of a transparent plane (60x60 cm, width x length), positioned 80 cm high above the floor. The half of the arena was immediately beneath the platform covered by a black-and-white checkered sheet (pixel size; 2x2 cm) (shallow side) and the other half was a transparent plane, below which the identical checkerboard pattern was located on the floor (deep side). Mouse behavior was recorded using the camera, and the percentage of time spent on each side during the first 5 min was calculated by an observer in a blind manner.

## Statistical analysis

In *Figures 2H–L, 3E, F and a* generalized linear mixed model (GLMM) was used. In *Figures 2F, G, 9D, F and a* log1p-transformation followed by a linear mixed model (LMM) was applied. In *Figures 2M and 9B*, *Figure 4—figure supplement 1A-C and E*, Student's t-test was used. In *Figure 8C and D*, Dunn's test was used. In *Figure 8F*, the one-way ANOVA followed by Tukey's multiple comparisons test was applied. In *Figure 3—figure supplement 1C*, LMM was used. In *Figures 5C–F , and 7D*, the Mann-Whitney U test was used. In *Figure 6A*, the Kolmogorov-Smirnov test was used. In *Figure 4—figure supplement 1D, F*, and *Figure 9—figure supplement 1B–D*, in which multiple stages and genes were analyzed, and data distribution and variance differed across conditions. Statistical tests (Student's t-test, Welch's t-test, or Mann–Whitney U test) were chosen accordingly. The specific statistical test used for each stage or gene is described in the corresponding Figure legends. Error bars denote standard error. The following asterisks in Figures indicate *p* values: *<0.05, **<0.01, ***<0.001.

## Acknowledgements

We thank Dr. Takahisa Furukawa for providing *Dkk3*-Cre mice, Dr. Toshihiko Fujimori for providing R26R-H2B-EGFP mice, Takefumi Yamamoto for technical assistance in electron microscopy, Drs. Akishi Onishi, Katsunori Kitano, and Kiyo Sakagami for helpful comments, and Y Shibata for technical assistance. This work was supported by Grant-in-Aid for Scientific Research (B) (24390019), Fund for the Promotion of Joint International Research (22KK0137), Early-Career Scientists (23K15920), and Research Activity Start-up (22K20698) from the Japan Society for the Promotion of Science (JSPS), Takeda Science Foundation, the Kobayashi Foundation, JST PRESTO (08062795), and R-GIRO.

# Additional information

## Funding

| Funder | Grant reference number | Author |
|---|---|---|
| Japan Society for the Promotion of Science | JP24390019 | Chieko Koike |
| Japan Society for the Promotion of Science | JP22KK0137 | Chieko Koike |
| Japan Society for the Promotion of Science | JP23K15920 | Akiko Ueno |
| Japan Society for the Promotion of Science | JP22K20698 | Akiko Ueno |
| Takeda Science Foundation | | Chieko Koike |
| Kobayashi Foundation | | Chieko Koike |
| Japan Science and Technology Agency | 08062795 | Chieko Koike |
| Ritsumeikan Global Innovation Research Organization | | Chieko Koike |

The funders had no role in study design, data collection and interpretation, or the decision to submit the work for publication.

## Author contributions

Akiko Ueno, Conceptualization, Formal analysis, Funding acquisition, Investigation, Visualization, Methodology, Writing – original draft, Project administration, Writing - review and editing; Konan Sakuta, Formal analysis, Visualization, Methodology; Hiroki Ono, Investigation, Visualization, Methodology; Aki Hashio, Formal analysis, Investigation, Visualization, Methodology; Haruki Tokumoto, Mikiya Watanabe, Shunsuke Mizuno, Investigation; Taketo Nishimoto, Formal analysis, Visualization; Toru Konishi, Investigation, Visualization; Yuki Emori, Formal analysis; Mao Hiratsuka, Investigation, Methodology; Jun Miyoshi, Yoshimi Takai, Resources; Masao Tachibana, Conceptualization, Data curation, Formal analysis, Writing – original draft, Project administration, Writing - review and editing; Chieko Koike, Conceptualization, Supervision, Funding acquisition, Writing – original draft

## Author ORCIDs

Akiko Ueno (ID) https://orcid.org/0009-0005-7507-3559
Chieko Koike (ID) https://orcid.org/0000-0002-1927-017X

## Ethics

All animal experimental protocols were conducted in accordance with local guidelines and the ARVO statement on the use of animals in ophthalmic and vision research. These procedures were approved by the Institutional Safety Committee on Recombinant DNA Experiments (approval ID R4016) and Animal Experimental Committees (approval ID BKC2022-017) of Ritsumeikan University.

Reviewer #1 (Public review): https://doi.org/10.7554/eLife.105627.3.sa1
Reviewer #2 (Public review): https://doi.org/10.7554/eLife.105627.3.sa2
Author response https://doi.org/10.7554/eLife.105627.3.sa3

# Additional files

## Supplementary files

MDAR checklist

Supplementary file 1. The primer sequences used for RT-qPCR.

## Data availability

Source data for Figures 2F-M; 3E, F; 4D; 5A-F; 6C, D; 7B-H; 8C, D, F; 9B, D, F; 3-figure supplement 1C; 4-figure supplement 1A-F; and 9-figure supplement 1B-D, together with full-size blot images of Figure 2-figure supplement 1C, are available on Zenodo.

The following dataset was generated:

| Author(s) | Year | Dataset title | Dataset URL | Database and Identifier |
|---|---|---|---|---|
| Koike C, Ueno A, Sakuta K, Ono H, Hashio A | 2025 | Afadin-deficient retinas exhibit severe neuronal lamination defects but preserve visual functions | https://doi.org/10.5281/zenodo.17037918 | Zenodo, 10.5281/zenodo.17037918 |

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
