## [Editor Report · eLife Assessment]

This study demonstrates that conditional knockout of afadin disrupts retinal laminar organization and reduces the number of photoreceptors, while preserving certain aspects of retinal ganglion cell structure and light responsiveness. The work is **valuable** and well-supported by revised figures and comprehensive data on retinal cell types, lamination patterns, and visual functio. The findings are **solid** and intriguing, and the study provides insights into the relationship between retinal lamination and neural circuit function.

---

## [Referee Report · Reviewer #1 (Public review)]

Summary:

The question of how central nervous system lamination defects affect functional integrity is an interesting yet debated topic. The authors investigated the role of afadin, a key adherens junction scaffolding protein, in retinal lamination and function using a retina-specific conditional knockout mouse model. Their findings show that the loss of Afadin caused severe outer retinal lamination defects, disrupting photoreceptor morphology, synapse numbers, and cell positioning, as demonstrated by histological analysis. Despite these structural impairments, retinal function was partially preserved: mERG detected small a- and b-waves, retinal ganglion cells responded to light, and behavioral tests confirmed residual visual function. This research offers new insights into the relationship between retinal lamination and neural circuit function, suggesting that altered retinal morphology does not completely eliminate the capacity for visual information processing.

Strengths:

The study effectively employs the well-organized laminar structure of the retina as an accessible model for investigating afadin's role in lamination within the central nervous system. High-quality histological, immunostaining, and electron microscopy images clearly reveal structural defects in the conditional knockout mice. The revised manuscript significantly enhances the findings by incorporating robust quantitative analyses of cell positioning, retinal thickness, and cell numbers, as well as new assessments of developmental defects. Additionally, new behavioral tests, including the optomotor response and visual cliff tests, have been introduced. Together with electrophysiological recordings, these additions compellingly demonstrate the partial preservation of visual function despite severe structural disruptions.

Weaknesses:

Overall, the study of the mechanisms remains weak. While the authors addressed concerns about molecular mechanisms by examining cell proliferation potentially related to Notch and Wnt signaling (Figure S6C, lines 868-870), the findings are largely negative (no significant changes in progenitor cell numbers), and the discussion of alternative pathways remains speculative.

---

## [Referee Report · Reviewer #2 (Public review)]

Summary:

Ueno et al. described substantial changes in the Afadin knockout retina. These changes include decreased numbers of rods and cones, an increased number of bipolar cells, and disrupted somatic and synaptic organization of the outer limiting membrane, outer nuclear layer, outer plexiform layer. In contrast, the number and organization of amacrine cells and retinal ganglion cells remain relatively intact. They also observed changes in ERG responses, RGC receptive fields and functions, and visual behaviors. The morphological and function characterization of retinal cell types and laminations is detailed and relatively comprehensive.

---

## [Author Response]

The following is the authors’ response to the original reviews

**Reviewer #1 (Public review):**
Summary:The question of how central nervous system (CNS) lamination defects affect functional integrity is an interesting topic, though it remains a subject of debate. The authors focused on the retina, which is a relatively simple yet well-laminated tissue, to investigate the impact of afadin - a key component of adherens junctions on retinal structure and function. Their findings show that the loss of afadin leads to significant disruptions in outer retinal lamination, affecting the morphology and localization of photoreceptors and their synapses, as illustrated by high-quality images. Despite these severe changes, the study found that some functions of the retinal circuits, such as the ability to process light stimuli, could still be partially preserved. This research offers new insights into the relationship between retinal lamination and neural circuit function, suggesting that altered retinal morphology does not completely eliminate the capacity for visual information processing.Strengths:The retina serves as an excellent model for investigating lamination defects and functional integrity due to its relatively simple yet well-organized structure, along with the ease of analyzing visual function. The images depicting outer retinal lamination, as well as the morphology and localization of photoreceptors and their synapses, are clear and well-described. The paper is logically organized, progressing from structural defects to functional analysis. Additionally, the manuscript includes a comprehensive discussion of the findings and their implications.Weaknesses:While this work presents a wealth of descriptive data, it lacks quantification, which would help readers fully understand the findings and compare results with those from other studies. Furthermore, the molecular mechanisms underlying the defects caused by afadin deletion were not explored, leaving the role of afadin and its intracellular signaling pathways in retinal cells unclear. Finally, the study relied solely on electrophysiological recordings to demonstrate RGC function, which may not be robust enough to support the conclusions. Incorporating additional experiments, such as visual behavior tests, would strengthen the overall conclusions.

We would like to thank the reviewer for the thoughtful and valuable comments that helped us to further improve the manuscript. We have revised the manuscript to address the following three points in response to the reviewer's comments.

While this work presents a wealth of descriptive data, it lacks quantification, which would help readers fully understand the findings and compare results with those from other studies.

In response, we quantified the position of each retinal cell type and measured retinal thickness in the cHet and cKO mice at 1M, as presented in Figures 2F–M. To reflect these additions, we have included explanatory text in the revised manuscript (see lines 507–533).

Furthermore, the molecular mechanisms underlying the defects caused by afadin deletion were not explored, leaving the role of afadin and its intracellular signaling pathways in retinal cells unclear.

As AJ components, such as catenin and cadherin, are known to be associated with several signaling pathways, including Notch and Wnt signals (PMID: 37255594), we speculated that these pathways might be disrupted in the afadin cKO retina. Since these pathways are involved in cell proliferation, we examined the number of progenitor cells in the afadin cKO retina at developmental stages P1, P3, and P5 (new Figure S6C, see lines 868-870). No significant differences were observed at any of these stages. We also quantified the number of each retinal cell type at P14 when differentiation is complete. In the cKO retina, the number of BCs significantly increased, whereas the number of photoreceptors significantly reduced (new Figure S4C, see lines 620-622). To our knowledge, activation or inactivation of any AJ-associated signaling pathway does not reproduce the cell fate alterations observed in the afadin cKO retina. These findings suggest that the above pathways related to AJ may be unchanged in the cKO retina. However, we cannot exclude the possibility that multiple signaling pathways may be affected simultaneously or other pathways affected in the cKO retina.

Finally, the study relied solely on electrophysiological recordings to demonstrate RGC function, which may not be robust enough to support the conclusions. Incorporating additional experiments, such as visual behavior tests, would strengthen the overall conclusions.

We appreciate the reviewer’s insightful suggestion. To more robustly evaluate visual function in the cKO mice, we performed optomotor response (OMR) and visual cliff tests using cHet, cKO, and optic nerve crush (ONC) mice with Aki Hashio, Yuki Emori, and Mao Hiratsuka. We added their name as co-authors to the new manuscript. In the OMR test, cKO mice exhibited fewer responses to visual stimuli than cHet mice but significantly more than ONC mice. Furthermore, although no significant difference was detected between cKO and ONC mice in the visual cliff test, some cKO mice displayed cautious behavior suggestive of depth perception. These results indicate that cKO mice retain partial visual function, which is consistent with the MEA analysis. We have included these data as the new Figure 8 and incorporated the findings into the revised manuscript in the Introduction (lines 130-131 and 133-134), Methods (lines 378-406), Results (lines 775-816), and Discussion sections (lines 1026-1035).

**Reviewer #2 (Public review):**
Summary:Ueno et al. described substantial changes in the afadin knockout retina. These changes include decreased numbers of rods and cones, an increased number of bipolar cells, and disrupted somatic and synaptic organization of the outer limiting membrane, outer nuclear layer, and outer plexiform layer. In contrast, the number and organization of amacrine cells and retinal ganglion cells remain relatively intact. They also observed changes in ERG responses and RGC receptive fields and functions using MEA recordings.Strengths:The morphological characterization of retinal cell types and laminations is detailed and relatively comprehensive.Weaknesses:(1) The major weakness of this study, perhaps, is that its findings are predominantly descriptive and lack any mechanistic explanation. As afadin is key component of adherent junctions, its role in mediating retinal lamination has been reported previously (see PMCID: PMC6284407). Thus, a more detailed dissection of afadin's role in processes, such as progenitor generation, cell migration, or the formation of retinal lamination would provide greater insight into the defects caused by knocking out afadin.

Thank you for valuable comments. We agree with the reviewer's point that findings are predominantly descriptive and lack any mechanistic explanation. However, we would like to clarify that the study cited in the comment (PMCID: PMC6284407) analyzed the role of afadin in dendritic stratification of direction-selective RGCs within the IPL, where “lamination” refers to the layering of RGC dendrites in the IPL. Here, we analyzed the function of afadin in the laminar construction of the overall retina.

In response to the reviewer’s comment, we have added new analyses addressing retinal lamination, as well as the number and spatial distribution of progenitor cells, during development in the cKO retina. These new results are shown in Figures 4E, 9C–F, S5A–C, and S6C of the revised manuscript, and corresponding explanations added in the revised text (lines 643–662 and 855–870).

(2) The authors observed striking changes in the numbers of rods, cones, and BCs, but not in ACs or RGCs. The causes of these distinct changes in specific cell classes remain unclear. Detailed characterizations, such as the expression of afadin in early developing retina, tracing cell numbers across various early developmental time points, and staining of apoptotic markers in developing retinal cells, could help to distinguish between defects in cell generation and survival, providing a better understand of the underlying causes of these phenotypes.

Thank you for the insightful comment. Following the reviewer’s suggestion, we quantified the number of retinal cell types at P14 when cell differentiation is complete (new Figure S4C). At P14, the numbers of photoreceptors and BCs were significantly reduced in the cKO retina, while Müller glia, which was significantly reduced at 1M, showed no difference. We further examined the number of rods and BCs at P1, P3, and P5 (new Figures S4E, F). No significant differences were detected at P1 or P3, however, at P5, rod marker expression was significantly decreased, while the number of BCs was significantly increased. These results suggest that the defects in cell fate determination of BCs and rods begin to emerge between P3 and P5, a period for which rods and BCs actively differentiate. We speculate that cells originally destined to become rods may instead differentiate into BCs in the cKO retina. In addition, we found a significant increase in apoptotic cells at P1, P3, P5, and P14 (new Figure S6B). Furthermore, Müller glia and rod photoreceptors showed significantly greater reduction at 1M compared to P14, suggesting that the reduction in Müller glia observed at 1M may be due to post-differentiation cell death. These are presented in Figures S4C, S4E–F, and S6B, and described in the revised manuscript (lines 620-635 and 827-838).

(3) Although the total number of ACs or RGCs remains unchanged, their localizations are somewhat altered (Figures 2E and 4E). Again, the cause of the altered somatic localization in ACs and RGCs is unclear.

Thank you for the valuable question. In response to the reviewer’s comment, we analyzed the position of RGCs and ACs in the developing cKO retina. In the cKO retina at P1, retinal cells were organized into distinct multicellular compartments with clear boundaries, and acellular regions extending to the outer retinal surface were observed at these boundaries. These acellular regions contained dendritic processes of RGCs and ACs, which are components of the IPL, indicating that elements of the IPL extended vertically across the retina. As development progressed, the compartment boundaries gradually shifted toward the inner retina. At P14, the IPL was mainly located on the inner retina, as in the normal retina. However, some IPL structures remained in the outer retina and may correspond to the acellular patches. We have included the above data in the revised manuscript as Figures S5A and S5B and revised the manuscript to include this point (lines 643-660).

(4) One conclusion that the authors emphasise is that the function of RGCs remains detectable despite a major disrupted outer plexiform layer. However, the organization of the inner plexiform layer remains largely intact, and the axonal innervation of BCs remains unchanged. This could explain the function integrity of RGCs. In addition, the resolution of detecting RGCs by MEA is low, as they only detected 5 clusters in heterozygous animals. This represents an incomplete clustering of RGC functional types and does not provide a full picture of how functional RGC types are altered in the afadin knockout.

We appreciate the reviewer’s insightful comments. Although our clustering of RGC subtypes in afadin cHet retinas resulted in only five clusters, the key finding of our study is the preservation of RGC receptive fields in afadin cKO retinas, despite severe photoreceptor loss (reduced to about one-third of normal) and disruption of photoreceptor-bipolar cell synapses in the OPL. This suggests that even with crucial damage to the OPL, the primary photoreceptor-bipolar-RGC pathway can still function as long as the IPL remains intact. Moreover, the presence of rod-driven responses in RGCs indicates that the AII amacrine cell-mediated rod pathway may also continue to function. We agree that our functional clustering in afadin cHet retinas was incomplete. However, we guess that the absence of RGCs with fast temporal responses in afadin cKO retinas may not simply be due to the loss of specific RGC subtypes but due to disrupted synaptic connections between photoreceptors and fast-responding BCs. Furthermore, the structural abnormalities in retinal lamination in afadin cKO retinas may alter RGC response properties, making strict functional classification less meaningful. We would like to emphasize the finding that disruption of the retinal lamination in afadin cKO retinas leads to the absence of RGCs with fast temporal response properties, rather than focusing solely on the classification of RGC subtypes.

Minor Comments:(1) Line 56-67: "Overall, these findings provide the first evidence that retinal circuit function can be partially preserved even when there are significant disruptions in retinal lamination and photoreceptor synapses" There is existing evidence showing substantial adaption in retinal function when retinal lamination or photoreceptor synapses are disrupted, such as PMCID: PMC10133175.

Thank you for your comment. We agree that the original sentence was ambiguous in its wording, and we have revised it to clarify our intended meaning (lines 48-50):

"Overall, these findings provide the first evidence that retinal circuit function can be partially preserved even when there are significant disruptions in both retinal lamination and photoreceptor synapses."

In response, we have cited this study and added the following sentence to the Discussion section of the revised manuscript. The paper you mentioned is crucial for discussing and considering the results of our study. We have cited this study and added the following sentence to the Discussion section of the revised manuscript (lines 910-915):

“Furthermore, RFs of RGCs are also detected in several mouse models of retinitis pigmentosa, in which rod photoreceptors are degenerated and surviving cone photoreceptors lose their OS discs and pedicles, instead forming abnormal processes resembling synaptic dendrites (Barhoum et al., 2008; Ellis et al., 2023; Scalabrino et al., 2022).”

(2) Line 114-115: "we focused on afadin, which is a scaffolding protein for nectin and has no ortholog in mice." The term "Ortholog" is misused here, as the mouse has an afadin gene. Should the intended meaning be that afadin has no other isoforms in mouse?

Thank you for pointing it out. As we misused "Ortholog" as "Paralog", we revised the sentence (line 108).

**Recommendations for the authors:**
(1) The introduction to afadin is insufficient. Please provide more background information about this protein.

Following the reviewer’s recommendations, we expanded the Introduction in the revised manuscript to provide a more detailed background on afadin, as follows (lines 108-119):

“Afadin regulates the localization of nectin, which initiates cell–cell adhesion and promotes AJ formation by recruiting the cadherin–catenin complex. (Ohama et al., 2018; Takai and Nakanishi, 2003). In addition, afadin interacts with various cell adhesion and signaling molecules, as well as the actin cytoskeleton, and contributes to the accumulation of β-catenin, αE-catenin, and E-cadherin at AJs (Sakakibara et al., 2018; Sato et al., 2006). Afadin KO mice exhibit severe disruption of AJs in the ectoderm, along with other developmental defects, leading to embryonic lethality (Ikeda et al., 1999; Zhadanov et al., 1999). Conditional deletion of afadin in RGCs leads to disruption of dendrites in ON-OFF direction-selective RGCs (Duan et al., 2018). However, the effect of afadin loss on retinal lamination, circuit formation, and function is poorly understood.”

(2) In Figure 1A (Bottom), regarding the peptide+ image, what does the green signal represent?

The green signal observed in the peptide+ image represents the background and non-specific staining. We have added the sentence to the legend of Figure 1A in the revised manuscript (lines 1067-1068).

(3) In the RESULTS section on page 17, the statement "Nectin-1, unlike nectin-2 and nectin-3, was partially co-localized with afadin at the OPL and IPL, in addition to the OLM" suggests that nectin-2 is also expressed at the IPL, as shown in Figure S1A. Providing high-power images, similar to those in Figure S1B, could help readers clearly recognize the staining signals.

Following your suggestion, we added higher-magnification images of Nectin-2 signals in the IPL to Figure S1A and included the following clarification in the Figure legend (lines 1356-1358):

“Nectin-2 and nectin-3 were localized in the OLM. The Nectin-2 signal in the IPL was insufficient for reliable assessment of its localization and colocalization.”

(4) Figure S2A requires an uncropped scan of the membrane after Western blotting to demonstrate that there are no non-specific bands when using this afadin antibody, which was also utilized for IHC.

We revised the new Figure S2C to include the uncropped membrane scan. Faint non-specific bands were observed in the Western blot, consistent with detecting non-specific signals in immunostaining using the anti-afadin antibody pre-absorbed with its antigen peptide.

(5) IHC staining is necessary to demonstrate the knockout of afadin in retinal cells, as the paper does not show Cre expression in the retinal cells of the Dkk3-Cre mouse line. This would also help verify the specificity of the afadin antibody.

In the cKO retina, the laminar structure was disrupted, and the background signal was generally high, making it difficult to reliably assess whether afadin expression was lost using immunostaining with the anti-afadin antibody. Therefore, in addition to the Western blot analysis already presented, we evaluated Cre activity in the Dkk3-Cre mouse line by crossing it with the R26-H2B-EGFP reporter line. Cre-mediated recombination was observed in all retinal cells at P0 and 1M. We have added these results to a revised Figure S2A and B and included explanatory text in the revised manuscript (lines 455–458).

(6) Why is the outer nuclear layer (ONL) severely impaired in the cKO mice when afadin is not expressed in this layer? Additionally, given that afadin is highly expressed in the inner plexiform layer (IPL), why does the cKO not affect its structure?

We speculate that the AJ defect in the outer retina during development may cause severe disruption of the ONL in afadin cKO mice. As shown in new Figure 9, ectopic AJs and aberrant position of mitotic cells were observed in the P0 cKO retina. These defects caused abnormal cell migration and position, resulting in the ONL disruption. On the other hand, in the IPL, afadin and other cell adhesion molecules may function redundantly, and thus, the IPL structure would be kept intact in the afadin cKO retina. We have added this interpretation to the Discussion section of the revised manuscript (lines 998–1005).

(7) In the RESULTS section on page 20, the authors state, "We further investigated adherens junctions (AJs) in the cKO retina by immunostaining with OLM adherens junction markers β-catenin, N-cadherin, and nectin-1. We found that these signals were dispersed in the cKO retina (Figure S2C)." It appears that β-catenin, N-cadherin, and nectin-1 can still be detected in the cKO retina.

We agree with the reviewer that β-catenin, N-cadherin, and nectin-1 can still be detected in the cKO retina. We used the term 'dispersed' to indicate that the signal was “scattered” rather than “disappeared”. To avoid confusion, we have revised the wording in the revised manuscript (line 499).\

(8) In Figure 3, please indicate where the zoomed-in images were captured from the low-power images. Additionally, point out the locations of zoomed-in images in other figures as well.

Following the reviewer’s suggestion, we updated Figures 2D, 3A-C, 4A, S2D, S3A, S3D, S3E, and S5D. The related Figure legends have also been revised.

(9) The authors should include individual data points in all statistical graphics to provide a clearer presentation of the data.

As suggested by the reviewers, we have revised all statistical graphs to display individual data points. Furthermore, the statistical analysis of synapse counts in Figures 3E, 3F, and S3C has been changed to linear mixed models (LMM) or generalized LMM to account for the variability in the number of synapses within individual mice.

(10) In the RESULTS section on page 23, the statement "These data indicate that the rosette-like structure in the cKO may be an ectopic IPL, termed 'acellular patches'". What is the mechanism that may cause the rosette-like structure to translocate from the IPL to the outer region of the retina?

Thank you for raising a valuable question. To clarify the mechanism of acellular patch formation in the cKO mice, we analyzed the position of RGCs and ACs in the developing cKO retina. In the cKO retina at P1, retinal cells were organized into distinct multicellular compartments with clear boundaries, and acellular regions extending to the outer retinal surface were observed at these boundaries. These acellular regions contained dendritic processes of RGCs and ACs, which are components of the IPL, indicating that elements of the IPL extended vertically across the retina. As development progressed, the compartment boundaries gradually shifted toward the inner retina. At P14, the IPL was mainly located on the inner retina, as in the normal retina. However, some IPL structures remained in the outer retina and may correspond to the acellular patches. We have included these findings in the revised manuscript as Figures S5A and S5B and added the corresponding description to the text (lines 643–665).

(11) Is the blood vessel structure normal in the cKO retina? Could this impact the survival of retinal cells?

Thank you for your valuable comment. We performed immunostaining with an anti-CD31 antibody, a marker for blood vessels, as shown in the new Figure S2G. No apparent differences were observed in the cKO retina. We have added the following description to the revised manuscript (lines 539–543):

“It has been reported that defects in the distal processes of Müller glia are associated with abnormal retinal vasculature (Shen et al., 2012). Thus, we immunostained the cKO retina with anti-CD31, a blood vessel marker, but no apparent vascular abnormalities were detected (Figure S2G).”

(12) In the RESULTS section on pages 26-29, there is a lot of statistical information included in parentheses. It would be more concise to place this information in the figure legends, if possible.

Following the reviewer's suggestion, we have moved the statistical information from the main text (pages 26–29) to the corresponding Figure legends.

(13) In the RESULTS section on page 28, the authors state, "On the other hand, the inner retina was apparently normal, and both the inner nuclear layer (INL) and IPL could be recognized." However, in Fig 7A, it appears that the INL is mixed with the ONL and cannot be clearly identified.

We agree with the reviewer that the INL is mixed with the ONL and cannot be clearly identified. Accordingly, we have revised the description in the text (lines 740–742) as follows:

“On the other hand, the inner retina was apparently normal, and both the IPL and the proximal part of the INL could be recognized.”.

(14) It is mentioned in the manuscript that "The receptive field (RF) area in the cKO retinas was significantly smaller than that in the cHet retinas." Is there an impairment in the dendritic fields of RGCs in the cKO retina that could lead to a smaller RF?

Thank you for asking an interesting question. The dendritic field reflects the region where presynaptic cells can form synaptic contacts, whereas the receptive field is dynamically shaped by spatiotemporal excitatory and inhibitory inputs, gap junctions, and membrane properties of the dendrites. Consequently, the size of the dendritic field does not necessarily correspond to that of the receptive field. Moreover, the disruption of the retinal lamination in the afadin cKO retina may alter the morphology of RGC dendritic fields—even when RNA expression levels are identical—which makes it difficult to exactly compare the morphology of the same RGC subtype between afadin cHet and afadin cKO retinas. Additionally, due to the presence of over 40 RGC subtypes and the rosette-like structures in the afadin cKO retina, it is challenging to trace the complete dendritic arborization of individual RGCs. For these reasons, we rather hesitate to compare the dendritic field size and the receptive field size.

(15) Figure 7H was not cited in the corresponding section of the main text.

Thank you for pointing it out. We have added a citation of Figure 7H in the revised manuscript (line 759).

(16) In Figure 8C, is there a difference in the number of pHH3+ mitotic cells between the cHet and cKO mice?

We quantified the number of pHH3-positive cells in the cKO retina at P0, as shown in the new Figure 9B. The number of mitotic cells was significantly increased in the cKO retina (see lines 853-855). In contrast, the number of BrdU-labeled progenitor cells at P1, P3, and P5 was not significantly different between cHet and cKO retinas, as presented in the new Figure S6C. These results suggest that although the total number of progenitor cells remain unchanged in cKO retinas, the M phase may be prolonged.

(17) The results related to Figure 8 should be moved to a location before Figure 5, as Figure 8 is also related to the lamination defects.

In the original manuscript, Figures 2–7 presented the phenotypes observed in the cKO retina, while Figure 8 addressed the possible cause of the lamination defects. Since the revised Figure 8 presents behavioral tests evaluating visual function, the phenotypic analyses are presented in the revised Figures 2–8. In response to the reviewers’ comments, we further analyzed the distribution of mitotic and progenitor cells during development and included these results as revised Figure 9.

(18) In the DISCUSSION section on page 32, the authors state, "A few photoreceptor-bipolar cell-retinal ganglion cell (BC-RGC) pathways (vertical pathways of the retina) are inferred to be maintained in the cKO retina." The authors could verify this using retrograde transsynaptic tracing with a pseudorabies virus injected into the superior colliculus.

Thank you for your interesting suggestion. This is an important point, and the recommended experiment idea sounds excellent. We attempted this analysis; however, the virus injected into the superior colliculus successfully labeled RGCs but failed to reach BCs and photoreceptors in normal mice. We guess that light stimulation evoked RGC firings evidently show that the photoreceptor-bipolar cell-retinal ganglion cell (BC-RGC) pathways function.